# ADAPTIVE DUAL PROMPTING: HIERARCHICAL DEBIASING FOR FAIRNESS-AWARE GRAPH NEURAL NETWORKS

## ABSTRACT

In recent years, pre-training Graph Neural Networks (GNNs) through self-supervised learning on unlabeled graph data has emerged as a widely adopted paradigm in graph learning. Although the paradigm is effective for pre-training powerful GNN models, the objective gap often exists between pre-training and downstream tasks. To bridge this gap, graph prompting adapts pre-trained GNN models to specific downstream tasks with extra learnable prompts while keeping the pre-trained GNN models frozen. As recent graph prompting methods largely focus on enhancing model utility on downstream tasks, they often overlook fairness concerns when designing prompts for adaptation. In fact, pre-trained GNN models will produce discriminative node representations across demographic subgroups, as downstream graph data inherently contains biases in both node attributes and graph structures. To address this issue, we propose an **A**daptive **D**ual **Prompt**ing (ADPrompt) framework that enhances fairness for adapting pre-trained GNN models to downstream tasks. To mitigate attribute bias, we design an Adaptive Feature Rectification module that learns customized attribute prompts to suppress sensitive information at the input layer, reducing bias at the source. Afterward, we propose an Adaptive Message Calibration module that generates structure prompts at each layer, which adjust the message from neighboring nodes to enable dynamic and soft calibration of the information flow. Finally, ADPrompt jointly optimizes the two prompting modules to adapt the pre-trained GNN while enhancing fairness. We conduct extensive experiments on four datasets with four pre-training strategies to evaluate the performance of ADPrompt. The results demonstrate that our proposed ADPrompt outperforms seven baseline methods on node classification tasks. Our code is available at:https://anonymous.4open.science/r/ADPrompt-18178.

## 1 INTRODUCTION

Graphs are ubiquitous in various real-world scenarios across diverse domains, including bioinformatics (Chatzianastasis et al., 2023), healthcare systems (Jiang et al., 2024), and fraud detection (Huang et al., 2022). Within the landscape of graph learning, Graph Neural Networks (GNNs) (Kipf & Welling, 2017; Hamilton et al., 2017a; Veličković et al., 2018a; Xu et al., 2019; Chen et al., 2020a; Rossi et al., 2020) are a preeminent paradigm, widely recognized for their remarkable capability to learn graph representations. In particular, GNN models leverage a message-passing mechanism (Kipf & Welling, 2017), wherein each node recursively aggregates information from its neighbors to obtain its node embedding. Traditionally, GNN models are optimized in an end-to-end manner for designated downstream tasks. However, this training paradigm critically depends on the availability of substantial labeled graph data, which is often limited in practical scenarios. Moreover, task-specific training often produces GNNs with limited generalization, restricting their applicability to other tasks.

To address the above issues, extensive research efforts have been devoted to developing effective graph pre-training strategies that leverage self-supervised learning to pre-train GNN models on unlabeled graph data (Veličković et al., 2019; Hu et al., 2020; You et al., 2020). When transferring pre-trained GNN models to specific downstream tasks, a primary challenge lies in the gap between

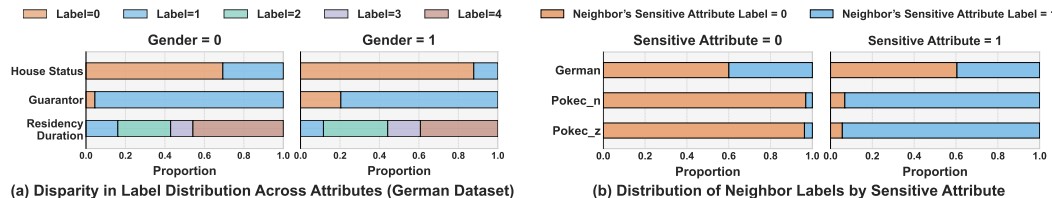

(a) Disparity in Label Distribution Across Attributes (German Dataset)    (b) Distribution of Neighbor Labels by Sensitive Attribute

Figure 1: (a) Attribute bias: In the German credit dataset, the distribution of labels varies across node attributes under different gender groups. (b) Structural bias: Across the three datasets, the distribution of sensitive attributes among node neighbors varies significantly across different sensitive groups, as defined in Table 3, indicating structural disparity.

the objectives of the pre-training and downstream tasks, e.g., link prediction during pre-training versus node classification in downstream tasks (Sun et al., 2022). Inspired by recent advances in prompting within computer vision and natural language processing (Jia et al., 2022; Zhou et al., 2022; Khattak et al., 2023; Yoo et al., 2023), graph prompting seeks to bridge this gap by adapting pre-trained GNN models for downstream tasks using additional tunable graph prompts (Sun et al., 2023b). In contrast to fine-tuning approaches that update the parameters of pre-trained GNN models for downstream tasks (Zhili et al., 2024; Sun et al., 2024; Huang et al., 2024), graph prompting modifies the input graph or its representations via learned prompts, while keeping the pre-trained model parameters fixed.

Although graph prompting effectively facilitates the adaptation of pre-trained GNN models to downstream tasks (Sun et al., 2022; Liu et al., 2023a; Fang et al., 2023; Yu et al., 2024b; Duan et al., 2024; Gong et al., 2024; Yu et al., 2024a; Li et al., 2025), existing graph prompting studies primarily focus on enhancing model utility (e.g., node classification accuracy), with limited attention to fairness concerns. These methods often overlook the potential for biased performance across demographic subgroups, such as gender and race. Beyond the inherent biases in pre-trained GNN models caused by the pre-training graph data and strategies, simply learning graph prompts on the biased graph data during adaptation can further exacerbate unfairness on downstream tasks. Unfortunately, such critical fairness concerns in graph prompting have not been fully explored yet, with only one recent study (Li et al., 2025) having undertaken a preliminary investigation into this crucial problem.

Debiasing pre-trained GNN models via graph prompting during adaptation is non-trivial due to two key challenges. The first challenge arises from *attribute bias*. For example, we can observe from Figure 1(a) that the distributions of three attributes (i.e., house status, guarantor, and residency duration) are inconsistent across different gender groups in the German credit dataset (Dua & Graff, 2017). In real-world scenarios, recruitment platforms may exhibit attribute bias, where user features like gender or age affect recommendations, limiting opportunities for some groups. In the graph data used for downstream tasks, sensitive information (e.g., gender or race information) is often carried by node attributes, both explicitly and implicitly. Intuitively, graph prompting can modify the graph data by directly masking these sensitive attributes to suppress their explicit influence on pre-trained GNN models. However, the remaining attributes can still encode sensitive information implicitly, thereby undermining efforts to promote fairness. The second challenge lies in *structure bias*, where graph connectivity patterns differ across demographic subgroups (Dai & Wang, 2021; Dong et al., 2022). For instance, in the Pokec_n and Pokec_z datasets (Figure 1(b)), neighbors' sensitive attributes vary markedly, with most nodes connected predominantly to same-group neighbors. Such patterns create "echo chambers" that limit exposure to diverse information. Through message passing, GNNs can further amplify these disparities: nodes with few neighbors receive limited information, and nodes surrounded by same-group neighbors receive homogeneous signals. These effects restrict information flow and reinforce biased representations, worsening unfairness in downstream tasks. As a result, these two challenges pose significant obstacles to achieving fair adaptation on downstream tasks. Notably, most existing work focuses on mitigating attribute bias, while structure bias has received comparatively little attention (Dai & Wang, 2021; Dong et al., 2023).

To overcome these challenges, we propose **A**daptive **D**ual **Prompt**ing (ADPrompt), a fairness-aware prompting framework, with soft and dynamic interventions. After pre-training a GNN, ADPrompt mitigates bias throughout message propagation in three complementary ways: (1) Adaptive Feature Rectification (AFR) purifies node attributes at the source by identifying and suppressing sensitive

feature dimensions; (2) Adaptive Message Calibration (AMC) dynamically adjusts messages between nodes across layers via edge-specific structure prompts; and (3) adversarial training encourages representations invariant to sensitive information. We present theoretical analyses highlighting that our lightweight prompting framework can both mitigate bias and enhance model adaptation on downstream tasks. Extensive experiments on four datasets and four pre-training strategies show that our method consistently outperforms seven baselines. Our contributions are summarized as follows:

- We propose a hierarchical fairness prompting framework spanning the entire information flow in GNNs, integrating source-level attribute purification with propagation-level message calibration. This dual-level design enables dynamic, soft adjustments that minimize disruption to the original graph structure, effectively mitigating bias while enhancing downstream performance.

- We present detailed theoretical analyses of ADPrompt, offering theoretical guarantees for both fairness and adaptability.

- We conduct extensive experiments on multiple datasets under four distinct GNN pre-training paradigms, and our results consistently show that our method outperforms seven representative baselines in both performance and fairness.

## 2 RELATED WORKS

### 2.1 GRAPH PROMPTING

Graph prompting has emerged as a paradigm to bridge pre-training and downstream tasks in GNNs, enabling efficient adaptation with minimal parameter updates (Liu et al., 2023b; Sun et al., 2023b; Dong et al., 2023). For instance, GPF and GPF-plus (Fang et al., 2023) propose a universal input-space framework that adapts diverse pre-trained GNNs without task-specific prompts. All in One (Sun et al., 2023a) presents a unified framework with learnable tokens and adaptive structures for flexible graph-level reformulation. GraphPrompt (Liu et al., 2023a) employs task-specific prompts to reweight node features during subgraph ReadOut, improving task-aware retrieval from frozen GNNs. TGPT (Wang et al., 2024) tailors prompts to graph topologies by leveraging graphlets and frequency embeddings for dynamic feature transformation. While graph prompting methods have shown promising progress in enhancing model performance and task adaptability, they generally fail to address the widespread bias inherent in real-world graph data (Dong et al., 2023).

### 2.2 GROUP FAIRNESS IN GRAPH LEARNING

As GNNs are increasingly applied across diverse domains, enhancing group fairness has become a critical focus in graph learning (Dai & Wang, 2021; Chen et al., 2024). The survey (Dong et al., 2023) reviews fairness in graph mining, proposes a taxonomy of fairness notions, and summarizes existing fairness techniques. FairDrop (Spinelli et al., 2021) proposes a biased edge dropout algorithm to counteract homophily and enhance fairness in graph representation learning. Graph counterfactual fairness (Ma et al., 2022) addresses biases induced by the sensitive attributes of neighboring nodes and their causal effects on node features and the graph structure. FPrompt (Li et al., 2025) enhances fairness in pre-trained GNNs with hybrid prompts that generate counterfactual data and guide structure-aware aggregation.

## 3 PRELIMINARIES

### 3.1 GRAPH NEURAL NETWORKS

Let $\mathcal{G} = (\mathcal{V}, \mathcal{E})$ denote an attributed graph with node set $\mathcal{V} = \{v_1, \ldots, v_N\}$ and edge set $\mathcal{E}$. Each node $v_i$ has an attribute vector $\mathbf{x}_i \in \mathbb{R}^{D_x}$, and $\mathcal{N}(v_i)$ denotes its neighbors. GNNs learn node representations via message passing (Kipf & Welling, 2017; Hamilton et al., 2017a; Veličković et al., 2018a), where each node iteratively aggregates information from its neighbors:

$$\mathbf{h}_i^{(l)} = \texttt{AGG}^{(l)}\Big(\mathbf{h}_i^{(l-1)}, \{\mathbf{h}_j^{(l-1)} : v_j \in \mathcal{N}(v_i)\}\Big), \tag{1}$$

with $\mathbf{h}_i^{(l)} \in \mathbb{R}^{D_l}$ the representation at layer $l$, initialized by $\mathbf{h}_i^{(0)} = \mathbf{x}_i$. The final embedding $\mathbf{h}_i^{(L)}$ is fed into a predictor $\pi$ for downstream tasks such as node classification.

## 3.2 FAIRNESS METRICS

In this study, we focus on group fairness (Dai & Wang, 2021; Dong et al., 2022), which requires models to produce non-discriminatory predictions across groups defined by sensitive attributes. Following prior work, we assess fairness using statistical parity (SP) and equal opportunity (EO), based on the binary label $y \in 0, 1$, sensitive attribute $s \in 0, 1$, and predicted label $\hat{y} \in 0, 1$. Specifically, the SP metric $\Delta$SP is defined as

$$\Delta\mathrm{SP} = |P\left(\hat{y} = 1 \mid s = 1\right) - P\left(\hat{y} = 1 \mid s = 0\right)| \tag{2}$$

and the equal opportunity (EO) metric $\Delta$EO is defined as

$$\Delta\mathrm{EO} = |P(\hat{y} = 1 \mid y = 1, s = 1) - P(\hat{y} = 1 \mid y = 1, s = 0)| \tag{3}$$

For both metrics, a lower value implies better fairness.

## 3.3 PROBLEM DEFINITION

Based on the above notations, we formulate the problem of fair graph prompting as follows.

**Problem 1.** *Given a graph $\mathcal{G} = (\mathcal{V}, \mathcal{E})$ and a GNN model $\theta^*$ pre-trained by a pre-training task $\mathcal{T}_{PT}$, fair graph prompting designs and learns extra tunable prompt vectors to adapt the pre-trained GNN model on a downstream task $\mathcal{T}_{DT}$ without tuning $\theta^*$. The goal of fair graph prompting is to improve model utility while enhancing model fairness.*

## 4 METHODOLOGY

Existing fairness methods for GNNs often suffer from limited adaptability: some rely on static augmentation (e.g., injecting fixed counterfactual prototypes), which ignores node-specific characteristics (Ma et al., 2022; Wo et al., 2025), while others adopt hard interventions (e.g., edge addition or deletion), which may disrupt critical topological information (Li et al., 2025).

To address these limitations, we propose Adaptive Dual Prompting (ADPrompt), a soft and dynamic intervention strategy that applies fine-grained adjustments to input node features and information flow while preserving the original graph structure. As illustrated in Figure 2, ADPrompt intervenes across the entire information flow in GNNs: it first purifies node attributes at the input layer via adaptive feature rectification, then calibrates message passing dynamically at each layer, and incorporates adversarial learning during optimization to enforce invariance to sensitive information. Details of each component are provided in the following subsections.

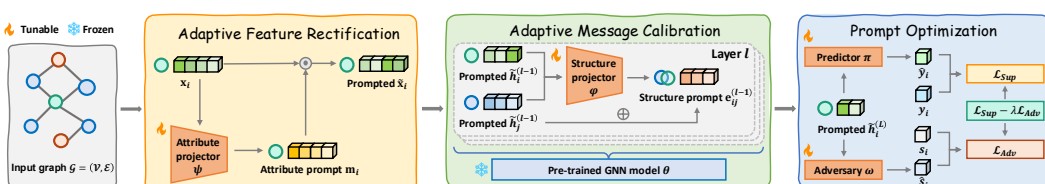

Figure 2: The framework of ADPrompt.

## 4.1 ADAPTIVE FEATURE RECTIFICATION

Attribute bias arises from sensitive information (e.g., gender or race) that is explicitly encoded in certain dimensions of node attributes and implicitly entangled with other dimensions. As a result, even if the explicit sensitive attributes are removed, biased outputs may still be produced by the pre-trained GNN. To counteract attribute bias during graph prompting, we introduce an Adaptive Feature

Rectification (AFR) module in ADPrompt. In this module, We purify information at the source level, reducing biased inputs from the outset. The intuition of AFR is to generate a personalized attribute prompt for each node that selectively attenuates sensitive feature dimensions via self-gating.

More specifically, each node $v_i \in \mathcal{V}$ will learn a customized attribute prompt $\mathbf{m}_i \in [0,1]^{D_x}$, which is then applied to its attribute vector $\mathbf{x}_i$ to obtain the prompted attribute vector $\tilde{\mathbf{x}}_i$ by

$$\tilde{\mathbf{x}}_i = \mathbf{m}_i \odot \mathbf{x}_i, \tag{4}$$

where $\odot$ represents the element-wise product between $\mathbf{m}_i$ and $\mathbf{x}_i$. However, learning $|\mathcal{V}|$ independent attribute prompts is impractical in graph prompting. During adaptation, only a small subset of nodes is typically labeled, so most nodes cannot receive supervision information for optimizing their attribute prompts. As a result, node $v_i$'s attribute prompt $\mathbf{m}_i$ cannot be reliably optimized if it never contributes to the representations of any labeled ones. To overcome this issue, we propose a self-gating mechanism (Hu et al., 2018) for Adaptive Feature Rectification. The intuition of the self-gating mechanism is to obtain attribute prompts for every nodes by learning a shared attribute projector $\psi$ followed by a sigmoid function. Here, the attribute projector $\psi$ is a compact network conditioned on the attribute vector of one node. More specifically, we compute the attribute prompt $\mathbf{m}_i$ for each node $v_i$ by

$$\mathbf{m}_i = \sigma(\psi(\mathbf{x}_i)) = \sigma(\mathtt{ReLU}(\mathbf{x}_i \mathbf{U}_1)\mathbf{U}_2), \tag{5}$$

where $\sigma$ is the sigmoid function. $\mathbf{U}_1 \in \mathbb{R}^{D_x \times D_u}$ and $\mathbf{U}_2 \in \mathbb{R}^{D_u \times D_x}$ are two learnable matrices in $\psi$. $D_u$ is one dimension size of $\mathbf{U}_1$ and $\mathbf{U}_2$. Then the prompted attribute $\tilde{\mathbf{x}}_i$ is subsequently used for the pre-trained GNN model. In Adaptive Feature Rectification, attribute prompts serve as input-layer gates that filter biased information from node features and mitigate bias at the source.

## 4.2 Adaptive Message Calibration

Although node features are purified at the outset, bias can still be amplified within GNNs due to structural disparities. Such disparities across demographic subgroups propagate through message passing, resulting in biased node representations. To address this, we introduce Adaptive Message Calibration (AMC) in ADPrompt. Our approach performs soft, fine-grained corrections at the information flow level without altering the graph structure. Hard interventions such as deleting or adding edges (Loveland et al., 2022; Franco et al., 2024; Li et al., 2025) rigidly modify the graph topology by removing existing connections or creating new ones. These modifications can alter the original relational patterns such as follower links, user interactions, and information flow in social networks. In contrast, AMC keeps the original topology unchanged and lets each node adjust how it receives and integrates information from its neighbors through learnable structure prompts.

More specifically, each node $v_i \in \mathcal{V}$ aims to learn a customized structure prompt $\mathbf{e}_{ij}^{(l-1)} \in \mathbb{R}^{D_{l-1}}$ that will be transmitted from $v_i$'s neighbor $v_j \in \mathcal{N}(v_i)$ along with edge $(v_i, v_j)$ at the $l$-th layer of the pre-trained GNN model. Mathematically, ADPrompt updates the prompted representation of node $v_i$ at the $l$-th layer by reformulating equation 1 with structure prompts as

$$\tilde{\mathbf{h}}_i^{(l-1)} = \mathtt{AGG}^{(l)} \left( \tilde{\mathbf{h}}_i^{(l-1)}, \left\{ \tilde{\mathbf{h}}_j^{(l-1)} + \mathbf{e}_{ij}^{(l-1)} : v_j \in \mathcal{N}(v_i) \right\} \right), \tag{6}$$

where $\tilde{\mathbf{h}}_i^{(0)} = \tilde{\mathbf{x}}_i$. Since $\mathbf{e}_{ij}^{(l-1)}$ depicts how the information transmitted from $v_j$ to $v_i$ at the $l$-th layer, we may naturally relate structure prompt $\mathbf{e}_{ij}^{(l-1)}$ to both $v_j$ and $v_i$. Considering this, we design a structure projector $\varphi$ to compute $\mathbf{e}_{ij}^{(l-1)}$ based on $\tilde{\mathbf{h}}_i^{(l-1)}$ and $\tilde{\mathbf{h}}_j^{(l-1)}$. Mathematically, the structure projector $\varphi$ computes $\mathbf{e}_{ij}^{(l-1)}$ by

$$\begin{aligned} \mathbf{e}_{ij}^{(l-1)} &= \varphi \left( \tilde{\mathbf{h}}_i^{(l-1)}, \tilde{\mathbf{h}}_j^{(l-1)} \right) \\ &= \mathtt{LeakyReLU} \left( \left[ \tilde{\mathbf{h}}_i^{(l-1)} \middle\| \tilde{\mathbf{h}}_j^{(l-1)} \right] \mathbf{W}_1^{(l)} \right) \mathbf{W}_2^{(l)}, \end{aligned} \tag{7}$$

where $[\cdot||\cdot]$ represents the vector concatenation. $\mathbf{W}_1^{(l)} \in \mathbb{R}^{2D_{l-1} \times D_w}$ and $\mathbf{W}_2^{(l)} \in \mathbb{R}^{D_w \times D_{l-1}}$ are learnable parameters in $\varphi$. $D_w$ is one dimension size of $\mathbf{W}_1^{(l)}$ and $\mathbf{W}_2^{(l)}$. Through Adaptive Message Calibration, ADPrompt dynamically adjusts the information flow during message passing, enabling layer-wise and fine-grained modifications based on current node embeddings.

## 4.3 PROMPT OPTIMIZATION

Through our Adaptive Feature Rectification and Adaptive Message Calibration, ADPrompt produces the final prompted representation $\tilde{\mathbf{h}}_i^{(L)}$ of node $v_i$. Ideally, $\tilde{\mathbf{h}}_i^{(L)}$ should be unbiased across demographic subgroups to ensure fair prediction for node classification. To achieve this, we design a joint optimization objective consisting of a supervised loss and an adversarial loss to collaboratively optimize attribute prompts and structure prompts.

### SUPERVISED LOSS.

During adaptation, the primary goal of ADPrompt is to enhance model utility on downstream tasks. In ADPrompt, the predictor $\pi$ uses $\tilde{\mathbf{h}}_i^{(L)}$ to generate node $v_i$'s prediction $\hat{y}_i = \pi(\tilde{\mathbf{h}}_i^{(L)})$. Given the labeled node set $\mathcal{V}_L \subset \mathcal{V}$, $\hat{y}_i$ is then used to predict the binary label $y_i$ of node $v_i$ by minimizing a supervised loss between $\hat{y}_i$ and $y_i$ for each labeled node $v_i \in \mathcal{V}_L$. Mathematically, we can formulate the supervised loss as the cross-entropy loss by

$$\mathcal{L}_{Sup}(\psi, \varphi, \pi) = -\frac{1}{|\mathcal{V}_L|} \sum_{v_i \in \mathcal{V}_L} [y_i \log \hat{y}_i + (1 - y_i) \log(1 - \hat{y}_i)]. \tag{8}$$

### ADVERSARIAL LOSS.

In the meantime, ADPrompt enhances fairness by encouraging prompted representations to be independent of sensitive attributes. To this end, we introduce a linear adversary $\omega$ that predicts each node's binary sensitive attribute from $\tilde{\mathbf{h}}_i^{(L)}$. By minimizing the adversary's predictive power, ADPrompt reduces sensitive information in the representations and guides the model toward fairer downstream predictions. More specifically, the adversary $\omega$ generates the predicted sensitive attribute $\hat{s}_i = \omega(\tilde{\mathbf{h}}_i^{(L)})$ for each node $v_i \in \mathcal{V}$. Given node $v_i$'s binary sensitive attribute $y_i$, we can formulate the adversarial loss as the cross-entropy loss by

$$\mathcal{L}_{Adv}(\psi, \varphi, \omega) = -\frac{1}{|\mathcal{V}|} \sum_{v_i \in \mathcal{V}} [s_i \log \hat{s}_i + (1 - s_i) \log(1 - \hat{s}_i)]. \tag{9}$$

### JOINT OPTIMIZATION OBJECTIVE.

By combining the above loss terms, we finally provide the joint optimization objective in ADPrompt as a minmax problem. Mathematically, the final optimization objective can be written as

$$\min_{\psi, \varphi, \pi} \max_{\omega} \mathcal{L}_{Sup}(\psi, \varphi, \pi) - \lambda \mathcal{L}_{Adv}(\psi, \varphi, \omega), \tag{10}$$

where $\lambda$ is a hyperparameter to balance the two loss terms. The complete algorithm of prompt optimization is provided in 1.

## 5 THEORETICAL ANALYSIS

In this section, we present a theoretical analysis to elucidate how the ADPrompt framework systematically mitigates group bias. Our analysis focuses on decomposing the upper bound of the Generalized Statistical Parity ($\Delta_{GSP}$) and showing how our dual prompting mechanism tightens this bound by alleviating its key contributing terms (Dai & Wang, 2021; Li et al., 2025). We further analyze model adaptability in Appendix A and provide an Information-Theoretic perspective in Appendix B.

### 5.1 PRELIMINARIES AND ANALYTICAL FRAMEWORK

**Fairness Criterion.** We adopt the Generalized Statistical Parity ($\Delta_{GSP}$) as our core fairness metric (Zafar et al., 2017). For models with continuous outputs, $\Delta_{GSP}$ is defined as the norm of the difference between the expected predictions across sensitive groups:

$$\Delta_{GSP}(\hat{y}) = \left\| \mathbb{E}[\hat{y}_i \mid s_i = 0] - \mathbb{E}[\hat{y}_i \mid s_i = 1] \right\|, \tag{11}$$

where $\hat{y}_i$ is the prediction of node $v_i$ and $\|\cdot\|$ denotes the $\ell_2$ norm. A smaller $\Delta_{GSP}$ indicates greater model fairness.

**Assumption 1.** *The activation functions of the GNN backbone $\theta$ and the classifier $\pi$ are **Lipschitz continuous**, with constants $L_f, L_\pi > 0$ (Rockafellar & Wets, 1998; Bartlett et al., 2017).*

**Assumption 2.** *The transformed representation $\tilde{X}$ satisfies the Markov condition $s \to X \to \tilde{X}$ and is uniformly bounded, and $\|\tilde{x}\| \le B$ almost surely for some constant $B > 0$ (Cover, 1999; van der Vaart & Wellner, 2014).*

**Analytical Framework.** We aim to minimize the expected prediction disparity $\Delta_{GSP}(\hat{y})$ across demographic groups defined by a sensitive attribute $s \in \{0, 1\}$. Under assumption 1, this disparity admits an upper bound, which is jointly determined by two main sources: (1) Initial Feature Bias, when node attributes explicitly carry or implicitly embed sensitive information; (2) Bias Amplification during Propagation, where the message-passing mechanism of GNNs can exacerbate the initial bias layer by layer. Our ADPrompt framework tightens this upper bound by synergistically addressing the two critical sources of bias.

## 5.2 FAIRNESS BOUND OF ADPROMPT

**Theorem 1** (Fairness Guarantee of ADPrompt). *Consider an $L$-layer GNN under Assumption 1. Let $\Delta_{GSP}(\tilde{X})$ denote the initial feature bias after AFR. Then the final-layer disparity satisfies*

$$\Delta_{GSP}(\tilde{h}^{(L)}) \le \Big( \prod_{l=1}^{L} \tilde{\gamma}^{(l)} \Big) \Delta_{GSP}(\tilde{X}) + \sum_{l=1}^{L} \Big( \prod_{k=l+1}^{L} \tilde{\gamma}^{(k)} \Big) \tilde{\epsilon}^{(l)}. \tag{12}$$

*The first term quantifies the propagation of initial feature bias through GNN layers, with $\tilde{\gamma}^{(l)} \le \gamma^{(l)}$ representing the AMC-calibrated amplification factor that bounds how much each layer can increase the bias. The second term represents the cumulative structural bias residuals introduced during message passing, where $\tilde{\epsilon}^{(l)} \le \epsilon^{(l)}$ is the AMC-calibrated residual factor controlling the per-layer structural bias.*

**Proof 1. Reduction of Initial Bias via AFR.** Under Assumption 2, AFR generates rectified features $\tilde{x}_i$ via equation 4, ensuring

$$I(\tilde{X}; s) \le I(X; s), \tag{13}$$

which means $\tilde{X}$ carries less information about the sensitive attribute $s$, thereby reducing the initial feature bias. By Pinsker's inequality (Cover, 1999), lower mutual information implies that the conditional distributions of the two groups are closer in total variation distance:

$$\|P(\tilde{X} \mid s = 0) - P(\tilde{X} \mid s = 1)\|_{TV} \le \sqrt{\frac{1}{2} D_{\mathrm{KL}}\Big( P(\tilde{X} \mid s = 0) \,\Big\|\, P(\tilde{X} \mid s = 1) \Big)} \tag{14}$$

$$\le \sqrt{\frac{1}{2} I(\tilde{X}; s)}. \tag{}$$

When two conditional distributions become closer in total variation, their expectations also move closer. Therefore, the difference between the group-wise expected rectified features satisfies

$$\Delta_{GSP}(\tilde{X}) = \big\| \mathbb{E}[\tilde{X} \mid s = 0] - \mathbb{E}[\tilde{X} \mid s = 1] \big\| \le \Delta_{GSP}(X), \tag{15}$$

showing that AFR yields a provably fairer initial representation.

**Proof 2. Suppression of Bias Amplification via AMC.** Although feature purification reduces initial bias, message passing may still amplify disparities. Let $\Delta^{(l)}$ denote the group disparity at the $l$-th layer:

$$\Delta^{(l)} = \big| \mathbb{E}[h_i^{(l)} \mid s_i = 0] - \mathbb{E}[h_i^{(l)} \mid s_i = 1] \big|. \tag{16}$$

For clarity of analysis, we present the GNN layer update in its standard matrix form:

$$h^{(l)} = \sigma\left(\hat{A}\, h^{(l-1)} W^{(l)}\right),\tag{17}$$

where $\hat{A}$ is a normalized adjacency matrix, $\sigma$ is a Lipschitz continuous activation function, and $W^{(l)} \in \mathbb{R}^{d_{l-1} \times d_l}$ is the trainable weight matrix of the $l$-th layer. The layer-wise group disparity is defined in Equation 16.

Using Lipschitz continuity of $\sigma$ and the sub-multiplicativity of matrix norms, we have

$$\begin{aligned}
\Delta^{(l)} &= \left\| \mathbb{E}\Big[\sigma(\hat{A}h^{(l-1)}W^{(l)}) \mid 0\Big] - \mathbb{E}\Big[\sigma(\hat{A}h^{(l-1)}W^{(l)}) \mid 1\Big] \right\| \\
&\leq L_\sigma \left\| \hat{A}\big(\mathbb{E}[h^{(l-1)} \mid 0] - \mathbb{E}[h^{(l-1)} \mid 1]\big)W^{(l)} \right\| + \epsilon^{(l)} \\
&\leq \underbrace{L_\sigma \|\hat{A}\|\,\|W^{(l)}\|}_{\gamma^{(l)}} \Delta^{(l-1)} + \epsilon^{(l)},
\end{aligned}\tag{18}$$

where $\epsilon^{(l)}$ collects residual structural or nonlinearity-induced discrepancies. AMC attenuates amplification along sensitive directions, yielding calibrated factors $\tilde{\gamma}^{(l)} \leq \gamma^{(l)}$ and $\tilde{\epsilon}^{(l)} \leq \epsilon^{(l)}$. The recursive bound equation 18 can be unrolled layer by layer:

$$\begin{aligned}
\Delta^{(L)} &\leq \tilde{\gamma}^{(L)}\Delta^{(L-1)} + \tilde{\epsilon}^{(L)}, \\
&\leq \tilde{\gamma}^{(L)}\big(\tilde{\gamma}^{(L-1)}\Delta^{(L-2)} + \tilde{\epsilon}^{(L-1)}\big) + \tilde{\epsilon}^{(L)}, \\
&= \tilde{\gamma}^{(L)}\tilde{\gamma}^{(L-1)}\Delta^{(L-2)} + \tilde{\gamma}^{(L)}\tilde{\epsilon}^{(L-1)} + \tilde{\epsilon}^{(L)}, \\
&\cdots \\
&= \Big(\prod_{l=1}^{L} \tilde{\gamma}^{(l)}\Big)\Delta^{(0)} + \sum_{l=1}^{L}\Big(\prod_{k=l+1}^{L} \tilde{\gamma}^{(k)}\Big)\tilde{\epsilon}^{(l)},
\end{aligned}\tag{19}$$

where $\Delta^{(0)} = \Delta_{GSP}(\tilde{X})$ denotes the initial feature bias after AFR. This completes the proof of Theorem 1.

## 6 EXPERIMENTS

### 6.1 EXPERIMENT SETTINGS

**Datasets.**  We employ four real-world graph datasets from various domains to evaluate our framework: (1) Credit defaulter (Yeh & Lien, 2009), a financial graph where nodes represent customers and edges indicate similar credit behavior; (2) German credit (Dua & Graff, 2017), a credit dataset where individuals are nodes and edges are based on feature similarities; (3) Pokec_z and (4) Pokec_n (Dai & Wang, 2021), social network subgraphs from the Pokec platform where nodes are users and edges represent friendships. More detailed information about datasets are in Appendix D.1.

**Pre-training Strategies.**  To evaluate the compatibility of our method, we adopt four pre-training strategies. For contrastive learning, we use InfoMax (Veličković et al., 2019), GraphCL (You et al., 2020), and BGRL (Thakoor et al., 2021). For generative learning, we use GAE (Kipf & Welling, 2016), which reconstructs graph structure from encoded node features. Further details are provided in Appendix D.2.

**Baselines.**  We compare our method with five state-of-the-art graph prompting approaches: Graph-Prompt (Yu et al., 2024b), GPF and its variant GPF-plus (Fang et al., 2023), Self-pro (Gong et al., 2024), and FPrompt (Li et al., 2025). Additionally, we report the performance of a classifier trained without prompts (named as Classifier Only), as well as a variant enhanced with adversarial learning (named as Adversarial Learning). All baselines are evaluated on node classification, which serves as our downstream task. More information on these baselines can be found in Appendix D.3.

Table 1: 50-shot performance comparison of graph prompting methods under four pre-training strategies over four datasets (all values in %). The best-performing method is **bolded** and the runner-up underlined.

| Pre-training | Tuning | Credit | | | German | | | Pokec_n | | | Pokec_z | | |
|---|---|---|---|---|---|---|---|---|---|---|---|---|---|
| | | ACC (↑) | ΔEO (↓) | ΔSP (↓) | ACC (↑) | ΔEO (↓) | ΔSP (↓) | ACC (↑) | ΔEO (↓) | ΔSP (↓) | ACC (↑) | ΔEO (↓) | ΔSP (↓) |
| InfoMax | Classifier only | 54.19 | 3.80 | 2.03 | 58.50 | 8.50 | 7.29 | 70.22 | 3.96 | 1.56 | 70.13 | 1.19 | 6.12 |
| | Adversarial learning | **59.67** | 2.97 | 2.91 | 61.80 | **0.82** | 6.39 | 71.68 | 0.98 | 2.67 | 70.11 | 0.99 | 5.01 |
| | GraphPrompt | 49.38 | 2.67 | 3.29 | 61.50 | 2.21 | 2.51 | **73.94** | 0.65 | 2.15 | **73.63** | 1.51 | 0.97 |
| | GPF | 56.68 | 3.67 | 2.41 | 58.17 | 1.70 | 6.34 | 69.03 | 2.80 | 1.89 | 68.94 | 3.12 | 1.94 |
| | GPF+ | 55.03 | 1.82 | 2.22 | 59.33 | 5.58 | 4.02 | 69.42 | 2.27 | 1.81 | 68.48 | 1.75 | 2.07 |
| | Self-pro | 46.49 | 2.86 | 2.13 | 61.83 | 8.95 | 5.80 | 66.89 | **0.29** | 2.54 | 70.40 | 0.98 | 3.16 |
| | FPrompt | 55.99 | 1.69 | 1.55 | 52.50 | 4.22 | 5.11 | 71.19 | 3.05 | 1.59 | 68.83 | 3.03 | 0.82 |
| | ADPrompt | 58.14 | **1.08** | **1.48** | **64.33** | 1.18 | **1.32** | 69.89 | **0.22** | **1.44** | 70.48 | **0.92** | **0.70** |
| GraphCL | Classifier only | 59.10 | 4.38 | 4.37 | 64.50 | 4.52 | 2.63 | 70.42 | 2.10 | 3.84 | 71.25 | 5.48 | 10.00 |
| | Adversarial learning | 58.50 | 2.45 | 3.75 | 57.83 | 2.58 | 4.06 | 70.41 | **0.35** | 3.12 | 70.64 | 3.40 | 6.26 |
| | GraphPrompt | 57.19 | 0.93 | 1.67 | 60.17 | 6.44 | 5.63 | 64.61 | 4.45 | 6.50 | 62.22 | 1.24 | 1.11 |
| | GPF | 55.19 | 2.23 | 2.04 | 50.07 | 7.09 | 3.73 | 69.44 | 4.13 | 2.10 | **74.41** | 3.35 | 1.43 |
| | GPF+ | 58.43 | 2.53 | 2.54 | 56.83 | 4.78 | 5.07 | 72.85 | 1.28 | 2.17 | 74.24 | 5.20 | 2.17 |
| | Self-pro | 57.61 | 1.51 | 2.27 | 62.00 | 2.96 | 5.51 | 71.78 | 5.27 | 1.51 | 71.18 | 0.93 | 2.50 |
| | FPrompt | 54.83 | 4.10 | 3.93 | 58.67 | 3.47 | 3.73 | 72.55 | 0.54 | 2.29 | 70.37 | 1.01 | 4.21 |
| | ADPrompt | **59.86** | **0.91** | 1.54 | **65.50** | 2.14 | 2.62 | **76.05** | 1.27 | **0.58** | 70.33 | **0.89** | **1.04** |
| GAE | Classifier only | 55.78 | 2.25 | 3.29 | 55.50 | 5.72 | 7.09 | 70.63 | 2.40 | 2.08 | 69.97 | 2.37 | 7.08 |
| | Adversarial learning | 53.31 | 3.81 | 4.40 | 57.00 | 8.47 | 9.78 | 69.50 | 3.64 | 0.79 | 71.09 | 1.37 | 2.95 |
| | GraphPrompt | 62.93 | 1.59 | 3.08 | 59.17 | 3.37 | 3.06 | **73.92** | 2.15 | 4.56 | 73.31 | 4.13 | 0.91 |
| | GPF | 54.63 | 3.39 | 2.23 | 50.33 | 7.38 | 4.68 | 68.94 | 2.24 | 1.71 | 67.67 | 2.69 | 1.52 |
| | GPF+ | 57.30 | 2.30 | 1.51 | 51.00 | 6.50 | 6.97 | 69.63 | 2.13 | 1.77 | 70.05 | 2.60 | 2.20 |
| | Self-pro | 57.91 | 1.43 | 1.37 | 50.67 | 2.71 | 3.57 | 73.54 | 1.93 | 0.47 | 74.03 | 1.53 | 1.60 |
| | FPrompt | 57.74 | 3.07 | 3.70 | 61.67 | 6.13 | 4.57 | 71.93 | 4.89 | 0.58 | 68.40 | 1.58 | **0.69** |
| | ADPrompt | **64.86** | **1.37** | **1.28** | **62.17** | 2.22 | 2.32 | 73.62 | 1.90 | **0.44** | **75.09** | 1.25 | 0.84 |
| BGRL | Classifier only | 55.03 | 2.88 | 3.59 | 63.33 | 7.23 | 9.60 | 70.09 | 2.22 | 4.43 | 70.34 | 6.30 | 1.11 |
| | Adversarial learning | 52.12 | 3.70 | 3.32 | 60.17 | 5.20 | 3.71 | 69.87 | 1.77 | 2.39 | 70.54 | 5.44 | 8.79 |
| | GraphPrompt | 56.06 | 4.52 | 2.89 | 58.49 | 5.54 | 4.96 | 73.63 | 1.51 | 2.42 | **70.68** | 2.45 | 2.77 |
| | GPF | 56.86 | 3.07 | 2.92 | 60.25 | 6.07 | 3.53 | 70.09 | 2.24 | 1.95 | 67.93 | 2.18 | 2.42 |
| | GPF+ | 56.87 | 3.21 | **1.58** | 61.00 | 8.49 | 4.49 | 70.98 | 1.24 | **1.77** | 69.39 | 2.22 | 1.10 |
| | Self-pro | 53.29 | 2.62 | 2.41 | 47.00 | 5.74 | 9.20 | 70.40 | 0.96 | 3.16 | 62.53 | **0.85** | 2.40 |
| | FPrompt | 55.80 | 2.42 | 2.95 | 64.50 | 3.35 | 3.43 | 68.83 | 3.03 | 1.82 | 65.07 | 1.43 | 0.82 |
| | ADPrompt | **58.24** | 2.12 | 1.89 | **65.00** | 2.34 | 2.37 | **75.63** | **0.92** | 1.77 | 66.12 | 0.99 | **0.67** |

**Implementation Details.** In our experiments, we adopt a 2-layer GCN (Kipf & Welling, 2017) as the backbone for node classification tasks. During pre-training, the datasets are randomly split into training, validation, and test sets with a ratio of 60%, 20%, and 20%, respectively. The hidden layer size is set to 128. We employ the Adam optimizer (Kipf & Welling, 2017) with a learning rate of 0.001 for all methods. We train graph prompting for 300 epochs. The main experiments adopt a 50-shot setting, while results for the 10-shot setting are reported in Appendix E.1. All experiments are repeated three times with different random seeds, and the average performance is reported.

## 6.2 MAIN RESULTS

We compare ADPrompt with seven baseline methods on 50-shot node classification tasks across four datasets under four pre-training strategies (Table 1). Our method consistently achieves the best or highly competitive performance across various pre-training strategies. While many baselines attain strong accuracy, they show notable fairness deficiencies, indicating unequal treatment of demographic groups. In contrast, ADPrompt reduces bias while maintaining high task performance: for example, on the German dataset, it improves accuracy by 3% and simultaneously lowers both ΔEO and ΔSP by 2%.

## 6.3 ANALYSIS OF ADPROMPT

**Analysis of Adaptive Feature Rectification.** To evaluate the AFR module, we analyzed the prompt coefficients of each feature dimension. Lower coefficients indicate stronger suppression of the corresponding feature dimension. As shown in Figure 3, sensitive attributes (e.g., gender, age) consistently receive lower values than non-sensitive ones, confirming that attribute prompts effectively suppress sensitive information and mitigate feature-level bias.

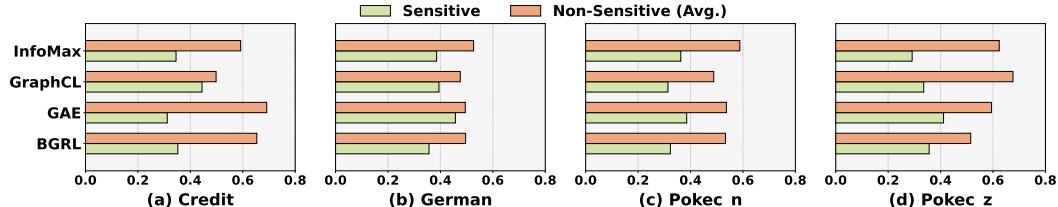

Figure 3: Comparison of prompt coefficients across sensitive and non-sensitive feature dimensions.

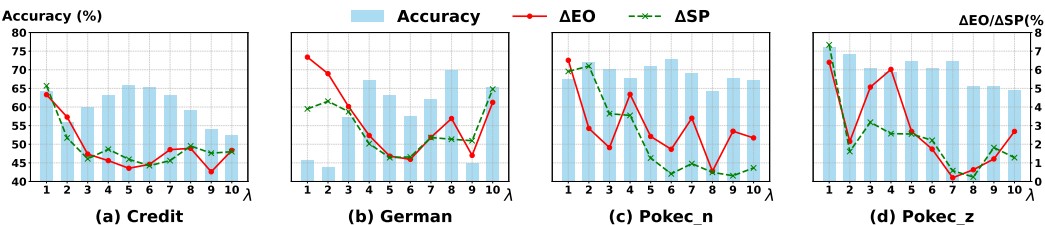

Figure 4: Effect of the hyperparameter $\lambda$ on accuracy and fairness under GraphCL pre-training.

**Impact of the Balancing Parameter.** To assess the trade-off between accuracy and fairness, we analyzed the impact of the balancing hyperparameter $\lambda$ in prompt optimization under the GraphCL pre-training strategy on four datasets. As shown in Figure 4, larger $\lambda$ increases the adversarial fairness loss $\mathcal{L}_{\text{Adv}}$, reducing fairness gaps but potentially harming accuracy. A moderate $\lambda$ (5–7) achieves the best balance across most datasets.

**Computational Efficiency Analysis.** Table 2 reports the running time and GPU memory usage of ADPrompt across four datasets under two pre-training paradigms. Each experiment was repeated three times, and the reported runtime and GPU usage represent the corresponding averages. The results indicate that the method is computationally lightweight: runtime remains consistently low across varying datasets, and the memory footprint stays within a modest range. Overall, these observations confirm that ADPrompt introduces minimal computational and memory demands, making it suitable for deployment on large-scale datasets.

Table 2: Running time (in seconds) of AD-Prompt. GPU indicates GPU memory usage.

| Dataset | Pre-training | Time (s) | GPU (GB) |
|---------|--------------|----------|----------|
| Credit  | InfoMax      | 6.19     | 1.08     |
|         | GraphCL      | 5.64     | 1.09     |
| German  | InfoMax      | 4.15     | 0.15     |
|         | GraphCL      | 3.88     | 0.11     |
| Pokec_n | InfoMax      | 64.04    | 8.26     |
|         | GraphCL      | 29.95    | 4.99     |
| Pokec_z | InfoMax      | 52.48    | 6.77     |
|         | GraphCL      | 30.12    | 4.14     |

**More experimental results.** Due to page limits, additional results are provided in Appendix E, including 10-shot performance comparison, multi-label classification tasks, computational efficiency comparisons, ablation studies, analyses with different backbones, and so on. These findings further demonstrate the effectiveness and robustness of ADPrompt across diverse settings.

## 7 CONCLUSION

This work presents ADPrompt, a fairness-aware graph prompting framework that jointly applies Adaptive Feature Rectification and Adaptive Message Calibration to mitigate bias in both node attributes and structural information. By introducing a small number of learnable prompts, ADPrompt effectively reduces group disparity in GNNs while enhancing adaptability to downstream tasks. Extensive experiments across multiple datasets and pre-training strategies demonstrate that ADPrompt consistently surpasses seven competitive baselines in both fairness and predictive performance. Looking ahead, we plan to investigate its scalability to large-scale graphs, its generalization across diverse graph modalities, and the design of more advanced prompting mechanisms to broaden its applicability.

## 8 REPRODUCTIVITY STATEMENT

To facilitate reproducibility, we release the complete implementation at `https://anonymous.4open.science/r/ADPrompt-18178`, along with the supplementary material, including pretrained GNN models and the core ADPrompt modules. All datasets used are publicly accessible, and additional experimental details are provided in Section 6.1 and Appendix D.

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

## A   THEORETICAL ANALYSIS OF MODEL ADAPTABILITY

In this section, we present a theoretical analysis of the adaptability of the ADPrompt framework. We demonstrate that ADPrompt can effectively adapt a fixed pre-trained GNN $\theta^*$ to diverse downstream tasks through the learned adaptive dual prompts. The key to this adaptability lies in the universality of the prompts, which can replicate any ideal modification of the graph structure and node attributes, thereby enabling near-optimal performance on the target task (Xu et al., 2018).

To formalize the adaptability of ADPrompt, let $\mathcal{G} = (\mathcal{V}, \mathcal{E})$ denote the original graph with node attributes $\mathbf{X} = [\mathbf{x}_1, \ldots, \mathbf{x}_N]^\top$. Let $\mathcal{G}' = (\mathcal{V}', \mathcal{E}')$ represent an arbitrary target graph from the candidate space, equipped with node attributes $\mathbf{X}' = [\mathbf{x}'_1, \ldots, \mathbf{x}'_N]^\top$. An ideal adaptation method would enable the pre-trained GNN $\theta^*$ to generate task-optimal node representations on $\mathcal{G}'$.

Based on this, we present the following theorem 2 to establish the universal adaptation capability of ADPrompt.

**Theorem 2** (Adaptability of ADPrompt). *Given a pre-trained L-layer GNN model $\theta^*$ and an input graph $\mathcal{G} = (\mathcal{V}, \mathcal{E})$ with node attributes $\mathbf{X}$, for any target graph $\mathcal{G}' = (\mathcal{V}', \mathcal{E}')$ with node attributes $\mathbf{X}'$, there exist learnable prompting modules $\psi$ (AFR) and $\varphi$ (AMC) such that the final node representations $\tilde{\mathbf{h}}_i^{(L)}$ produced by ADPrompt on $\mathcal{G}$ satisfy*

$$\tilde{\mathbf{h}}_i^{(L)}[\psi, \varphi] = \mathbf{h}_i'^{(L)}, \quad \forall v_i \in \mathcal{V}, \tag{20}$$

*where $\mathbf{h}_i'^{(L)}$ denotes the representation of node $v_i$ obtained by applying $\theta^*$ to the target graph $\mathcal{G}'$.*

The validity of Theorem 2 stems from the powerful expressiveness of the ADPrompt framework, which manifests in two key aspects:

**Proof 1. Simulating Arbitrary Feature Transformations via AFR.** The AFR module generates a personalized, dimension-wise attribute prompt $\mathbf{m}_i$ for each node $v_i$ and applies it via element-wise multiplication: $\tilde{\mathbf{x}}_i = \mathbf{m}_i \odot \mathbf{x}_i$. This fine-grained gating mechanism is significantly more expressive than simple additive prompts or global transformations. By optimizing the module $\psi$, the prompt $\mathbf{m}_i$ can be trained to arbitrarily scale, suppress, or even nullify each dimension of the original feature vector $\mathbf{x}_i$. This flexibility allows AFR to approximate any target feature matrix $\mathbf{X}'$ with high fidelity (Srivastava et al., 2015; Li & Liang, 2021; Ding et al., 2023).

**Proof 2. Simulating Arbitrary Structural Transformations via AMC.** Modifications to the graph structure (i.e., from $\mathcal{E}$ to $\mathcal{E}'$) fundamentally alter the message-passing pathways within the GNN (Franceschi et al., 2019; Chen et al., 2020b). The Adaptive Message Calibration (AMC) module directly intervenes in this process by injecting a layer-wise, edge-specific structure prompt $\mathbf{e}_{ij}^{(l-1)}$ into each message. This prompt can be learned to: (1) *strengthen or weaken* the message from a neighbor $v_j$ by aligning $\mathbf{e}_{ij}^{(l-1)}$ with $\tilde{\mathbf{h}}_j^{(l-1)}$, simulating changes in edge weights (Zhu et al., 2020); (2) *nullify* a message (e.g., when $\mathbf{e}_{ij}^{(l-1)} \approx -\tilde{\mathbf{h}}_j^{(l-1)}$), which is equivalent to removing the edge $(v_i, v_j)$; (3) *inject novel information* by designing $\mathbf{e}_{ij}^{(l-1)}$ independently of $\tilde{\mathbf{h}}_j^{(l-1)}$, simulating the effect of virtual nodes or edges present in the target graph $\mathcal{G}'$ but absent in the original graph. Because this intervention is layer-wise, edge-specific, and dynamic, AMC can effectively replicate the complex information flow resulting from any structural modification in $\mathcal{G}'$ (Xu et al., 2018).

In summary, the synergistic combination of AFR and AMC endows ADPrompt with expressive power to jointly simulate arbitrary feature and structural modifications of the input graph by learning the parameters of $\psi$ and $\varphi$. This establishes ADPrompt as a universal adapter that can guide the model to achieve a theoretical upper-bound of performance on downstream tasks.

## B   THEORETICAL ANALYSIS: FROM INFORMATION-THEORETIC PERSPECTIVE

We establish the theoretical foundation of ADPrompt's fairness capability through from information-theoretic perspective. In particular, we formulate fairness in graph representation learning as an **Information Bottleneck (IB)** problem (Tishby et al., 2000; Wu et al., 2020). The IB principle states

that an ideal node representation $\tilde{\mathbf{H}}$ should preserve maximal information about the task label $Y$, while suppressing information related to the sensitive attribute $S$. This trade-off can be formalized in the following Lagrangian form:

$$\min_{\theta_{ADPrompt}} I(\tilde{\mathbf{H}}; S) - \beta \, I(\tilde{\mathbf{H}}; Y), \tag{21}$$

where $I(\cdot; \cdot)$ denotes mutual information, $\theta_{ADPrompt}$ encompasses all learnable parameters (i.e., $\psi$ and $\varphi$), and $\beta > 0$ balances task relevance and sensitive information suppression. In practice, the training objective of ADPrompt, $\mathcal{L}_{Sup} - \lambda \mathcal{L}_{Adv}$, serves as an effective surrogate for this principle: $\mathcal{L}_{Sup}$ promotes the retention of task-relevant information $I(\tilde{\mathbf{H}}; Y)$, whereas $\mathcal{L}_{Adv}$ acts as a proxy for reducing sensitive information $I(\tilde{\mathbf{H}}; S)$.

**Proof 3. Adaptive Feature Rectification (AFR) as an Input-Layer Bottleneck.**   AFR acts as a bottleneck at the feature input layer (Tishby et al., 2000). Raw node attributes $\mathbf{X}$ often contain sensitive information correlated with $S$, forming the initial source of bias. By optimizing the projector $\psi$ adversarially, AFR generates a gating prompt $\mathbf{m}_i$ per node through equation 5. This element-wise gating selectively suppresses sensitive dimensions in $\mathbf{x}_i$, ensuring

$$I(\tilde{\mathbf{X}}; S) \leq I(\mathbf{X}; S), \tag{22}$$

and providing a purified feature foundation for fair downstream propagation (Jin et al., 2020).

**Proof 4. Adaptive Message Calibration (AMC) as a Layer-wise Regularizer.**   Input purification alone cannot prevent bias amplification in message passing (Dai & Wang, 2021; Dong et al., 2022). AMC serves as a layer-wise regularizer by generating edge-specific calibration vectors according to equation 7. These vectors act as corrective signals to suppress sensitive information propagated from neighbors, thereby reducing the mutual information:

$$I(\tilde{\mathbf{H}}_i^{(l)}; S_{\mathcal{N}(i)}) \leq I(\tilde{\mathbf{H}}_i^{(l-1)}; S_{\mathcal{N}(i)}), \tag{23}$$

which prevents bias accumulation across layers and ensures fairness in deep GNNs (Tishby et al., 2000; Moyer et al., 2018; Oono & Suzuki, 2019).

Together, AFR and AMC constitute a hierarchical information disentanglement strategy: (1) *Attribute-level:* AFR disentangles node features from sensitive attributes, thereby suppressing biased information leakage directly at the feature source. (2) *Structural-level:* AMC disentangles layer-wise representation updates from structural bias, acting as a progressive corrective mechanism that prevents the amplification of sensitive information throughout message passing. Trained under a unified adversarial objective, ADPrompt compresses sensitive information in the final node representations $\tilde{\mathbf{H}}$ while preserving task-relevant information about $Y$. This principled, information-theoretic design substantiates ADPrompt as an effective framework for fair graph representation learning.

## C   THE ALGORITHM OF ADPROMPT

The algorithm of ADPrompt is illustrated in Algorithm 1.

---

**Algorithm 1** ADPrompt

---

1: **Input:** pre-trained GNN model $\theta$; graph $\mathcal{G} = (\mathcal{V}, \mathcal{E})$ with node attributes $\mathbf{x}_i \in \mathbb{R}^{D_x}$ and neighbors $\mathcal{N}(v_i)$; hyperparameters: trade-off $\lambda$, learning rate $\eta$, total epochs $E$, current epoch $e$.
2: **Output:** attribute projector $\psi$, structure projector $\varphi$, predictor $\pi$,
3: **for** $e = 1$ to $E$ **do**
4:    **for** $v_i \in \mathcal{V}$ **do**
5:       Compute $m_i = \sigma(\psi(x_i))$ using equation 5
6:       Compute $\tilde{x}_i$ using equation 4
7:    **end for**
8:    **for** $l = 1$ to $L$ **do**
9:       **for** $v_i \in \mathcal{V}$ **do**
10:          **for** $v_j \in \mathcal{N}(v_i)$ **do**
11:             Compute $e_{ij}^{(l-1)} = \varphi(\tilde{h}_i^{(l-1)}, \tilde{h}_j^{(l-1)})$ using equation 7
12:          **end for**
13:          Update $\tilde{h}_i^{(l)}$ using equation 6
14:       **end for**
15:    **end for**
16:    **for** $v_i \in \mathcal{V}$ **do**
17:       Compute $\hat{y}_i = \pi(\tilde{h}_i^{(L)}), \quad \hat{s}_i = \omega(\tilde{h}_i^{(L)})$
18:    **end for**
19:    Compute $\mathcal{L}_{\text{Sup}}(\psi, \varphi, \pi)$ using equation 8
20:    Compute $\mathcal{L}_{\text{Adv}}(\psi, \varphi, \omega)$ using equation 9
21:    Update $\psi \leftarrow \psi - \eta \nabla_\psi [\mathcal{L}_{\text{Sup}}(\psi, \varphi, \pi) - \lambda \mathcal{L}_{\text{Adv}}(\psi, \varphi, \omega)]$
22:    Update $\varphi \leftarrow \varphi - \eta \nabla_\varphi [\mathcal{L}_{\text{Sup}}(\psi, \varphi, \pi) - \lambda \mathcal{L}_{\text{Adv}}(\psi, \varphi, \omega)]$
23:    Update $\pi \leftarrow \pi - \eta \nabla_\pi \mathcal{L}_{\text{Sup}}(\psi, \varphi, \pi)$
24:    Update $\omega \leftarrow \omega - \eta \nabla_\omega [\lambda \mathcal{L}_{\text{Adv}}(\psi, \varphi, \omega)]$
25: **end for**
26: **Return:** $\psi, \varphi, \pi$

---

## D    MORE DETAILS ABOUT EXPERIMENT SETUP

### D.1    INFORMATION ABOUT DATASET

Table 3 summarizes the statistics of the datasets used in our experiments. Each dataset is associated with a sensitive attribute (e.g., age, gender, or region) and a binary prediction target (e.g., default status, customer credibility). Notably, in the Pokec-z and Pokec-n datasets, the "working field" attribute has been binarized to facilitate binary classification tasks, as shown in prior work on fairness in graph learning. (Dai & Wang, 2021; Kose & Shen, 2024) For multi-class tasks, we consider datasets with labels of more than two categories. Specifically, Credit and German contain features with four classes each. In Pokec_n and Pokec_z, the age attribute is grouped by 0 to 18, 19 to 35, 36 to 55, and above 55, with roughly balanced numbers of nodes in each group, resulting in four-class labels. These datasets allow evaluation of model performance and fairness in multi-class prediction settings.

| Dataset | Nodes | Edges | Features | Sensitive | Binary Label | Multi-class Label | Classes |
|---------|-------|-------|----------|-----------|--------------|-------------------|---------|
| Credit | 30,000 | 1,421,858 | 13 | Age | Future default | Total Overdue Counts | 4 |
| German | 1,000 | 22,242 | 27 | Gender | GoodCustomer | Installment Rate | 4 |
| Pokec_z | 67,797 | 882,765 | 277 | Region | Working Field | Age | 4 |
| Pokec_n | 66,569 | 729,129 | 267 | Region | Working Field | Age | 4 |

Table 3: The statistics of the datasets used in our experiment.

### D.2    PRE-TRAINING STRATEGIES

We employ several representative graph pre-training methods as summarized below.

- **InfoMax** (Veličković et al., 2018b) is an unsupervised pre-training method that maximizes mutual information between node embeddings and a global summary via negative sampling, guiding the model to learn structure-aware representations for downstream tasks.
- **GraphCL** (You et al., 2020) is a contrastive learning-based approach that generates multiple structurally and semantically perturbed views of the same graph, resulting in robust and transferable node embeddings.
- **GAE** (Kipf & Welling, 2016) conducts self-supervised pre-training by encoding node features via a GCN and reconstructing the adjacency matrix through an inner product decoder, guiding the model to capture the graph's structural information.
- **BGRL** (Thakoor et al., 2021) employs a self-supervised learning paradigm where two augmented views of the same graph are processed by online and target encoders, and their representations are aligned using a bootstrapping loss.

### D.3 BASELINES

We evaluate our method against seven representative baselines. The details of each baseline are summarized as follows:

- **Classifier Only** is a non-prompting baseline that uses a basic classifier.
- **Adversarial Learning** is also a non-prompting baseline that enhances model robustness by training against perturbations to the graph structure or node features.
- **GraphPrompt** (Yu et al., 2024b) unifies pre-training and downstream tasks by introducing learnable prompt vectors into the readout layer of the graph encoder, which assists the model in retrieving task-relevant knowledge.
- **GPF** (Fang et al., 2023) is a universal graph prompt tuning method that operates in the input feature space. It achieves prompting by adding a shared, learnable vector to all node features, making it applicable to any pre-trained GNN.
- **GPF-plus** (Fang et al., 2023) improves upon GPF by using more sophisticated prompt designs in the input feature space to enhance performance on downstream tasks.
- **Self-pro** (Gong et al., 2024) handles heterophily by using an asymmetric graph contrastive learning framework, which generates prompts by structurally modifying the input graph to align pre-training and downstream objectives.
- **FPrompt** (Li et al., 2025) is a fairness-aware prompt tuning method that uses hybrid graph prompts to mitigate bias. It incorporates a fixed prompt to represent sensitive group embeddings and a learnable prompt to bridge the gap between pre-training and downstream tasks.

## E MORE EXPERIMENTAL RESULTS

### E.1 MODEL COMPARISON UNDER 10-SHOT SETTING

To assess the effectiveness of our model in few-shot scenarios, we conduct a comparative study against seven competitive baselines under the 10-shot setting. As shown in Table 4, our method consistently achieves superior performance in terms of both fairness and predictive accuracy. Each experiment is repeated three times, and the average results are reported.

### E.2 MODEL COMPARISON ON MULTI-LABEL CLASSIFICATION TASKS

Previous experiments focused on binary-label datasets. To further assess the generality of our method, we conducted experiments on multi-label classification tasks, with dataset details provided in Table 3 and results in Table 5. ADPrompt consistently demonstrates strong performance across all datasets, ranking among the top one or two in accuracy while maintaining low fairness disparities, and outperforming most baseline methods. These findings suggest that ADPrompt is effective beyond binary classification and generalizes well to multi-label settings.

Table 4: 10-shot performance comparison of graph prompting methods under four pre-training strategies over four datasets (all values in %).

| Pre-training | Tuning | Credit | | | German | | | Pokec_n | | | Pokec_z | | |
|---|---|---|---|---|---|---|---|---|---|---|---|---|---|
| | | ACC (↑) | ΔEO (↓) | ΔSP (↓) | ACC (↑) | ΔEO (↓) | ΔSP (↓) | ACC (↑) | ΔEO (↓) | ΔSP (↓) | ACC (↑) | ΔEO (↓) | ΔSP (↓) |
| InfoMax | Classifier only | 54.19 | 1.80 | 2.03 | 58.50 | 8.50 | 7.29 | 70.22 | 3.96 | 1.56 | 70.13 | 1.19 | 6.12 |
| | Adversarial learning | **59.67** | 2.97 | 2.91 | 61.80 | 0.82 | 6.39 | 71.68 | 0.98 | 2.67 | 70.11 | 0.99 | 5.01 |
| | GraphPrompt | 55.43 | 2.75 | 4.66 | 53.67 | 5.35 | 2.39 | 68.56 | 4.75 | 2.67 | 67.44 | 4.32 | 3.55 |
| | GPF | 57.39 | 3.87 | 2.81 | 56.33 | 3.77 | **1.07** | 68.60 | 2.83 | 1.46 | 66.11 | 1.31 | 3.65 |
| | GPF+ | 53.35 | 3.09 | 3.01 | 58.67 | 3.87 | 4.24 | 64.27 | 2.83 | 1.59 | 67.32 | 3.19 | **1.63** |
| | Self-pro | 58.93 | 1.61 | 2.19 | 61.32 | 4.07 | 6.94 | 71.60 | 2.04 | 4.97 | 69.94 | 6.83 | 5.02 |
| | FPrompt | 51.52 | 3.62 | 2.44 | 58.50 | **0.27** | 2.25 | 71.55 | 3.69 | 2.45 | 66.43 | 3.69 | 2.43 |
| | ADPrompt | 56.01 | **1.32** | **1.20** | **62.67** | 1.40 | 2.21 | **71.84** | **0.92** | **1.38** | **70.87** | **0.97** | 2.72 |
| GraphCL | Classifier only | 59.10 | 4.38 | 4.37 | **64.50** | 4.52 | 2.63 | 70.42 | 2.10 | 3.84 | 71.25 | 5.48 | 10.00 |
| | Adversarial learning | 58.50 | 2.45 | 3.75 | 57.83 | 2.14 | 4.06 | 70.41 | **0.35** | 3.12 | 70.64 | 3.40 | 6.26 |
| | GraphPrompt | 50.37 | 3.15 | 3.03 | 51.33 | 8.02 | 8.74 | 64.22 | 11.27 | 9.11 | 63.42 | 2.00 | 1.68 |
| | GPF | 56.76 | 2.19 | 1.89 | 53.50 | 9.19 | 8.97 | 69.29 | 6.57 | 6.44 | 63.40 | 6.24 | 7.80 |
| | GPF+ | 55.60 | 1.36 | 1.75 | 52.17 | 3.73 | 4.08 | **73.56** | 3.66 | 1.74 | 73.18 | 3.73 | 1.04 |
| | Self-pro | 52.71 | 1.92 | 2.13 | 64.00 | 5.44 | 4.74 | 70.85 | 6.99 | 6.80 | 68.87 | 1.19 | 1.86 |
| | FPrompt | 54.52 | 3.92 | 4.40 | 55.33 | 3.98 | 3.79 | 71.07 | 2.87 | 2.21 | 70.61 | 3.84 | 4.73 |
| | ADPrompt | **59.80** | **0.82** | **0.57** | 61.50 | **0.62** | 1.92 | 70.97 | 1.62 | 0.83 | 71.34 | 0.71 | 0.75 |
| GAE | Classifier only | 55.78 | 2.25 | 3.29 | 55.50 | 5.72 | 7.09 | 70.63 | 2.40 | 2.08 | 69.97 | 2.37 | 7.08 |
| | Adversarial learning | 53.31 | 3.81 | 4.40 | 57.00 | 8.47 | 9.78 | 69.50 | 3.64 | 0.79 | **71.09** | 1.37 | 2.94 |
| | GraphPrompt | 53.51 | 1.97 | 0.92 | 58.32 | 3.56 | 4.36 | 69.75 | 5.76 | 1.78 | 67.32 | 2.03 | 2.22 |
| | GPF | 51.12 | 1.99 | 1.77 | 53.50 | 4.99 | 5.73 | 66.41 | 1.33 | 1.32 | 67.24 | 1.88 | 2.33 |
| | GPF+ | 49.29 | 1.77 | 2.09 | 55.50 | 4.27 | 1.52 | **74.65** | 1.89 | 2.93 | 65.44 | 2.04 | 3.73 |
| | Self-pro | 46.98 | 2.70 | 1.60 | 45.00 | 5.01 | 5.01 | 64.03 | 2.53 | 4.58 | 66.44 | 3.88 | 8.74 |
| | FPrompt | 59.98 | 1.61 | 1.60 | 49.63 | 4.00 | 3.07 | 66.75 | 1.44 | 2.60 | 63.54 | 1.60 | 2.33 |
| | ADPrompt | **61.46** | 1.40 | 0.87 | **63.78** | **0.81** | 1.46 | 70.64 | **0.93** | **0.71** | 67.47 | **0.35** | **0.28** |
| BGRL | Classifier only | **55.03** | 2.88 | 3.59 | 63.33 | 7.23 | 9.60 | 70.09 | 2.22 | 4.43 | 70.34 | 6.30 | 1.11 |
| | Adversarial learning | 52.12 | 3.70 | 3.32 | 60.17 | 5.20 | 3.71 | 68.87 | 1.77 | 2.39 | 70.54 | 5.44 | 8.79 |
| | GraphPrompt | 54.62 | 4.44 | 4.70 | 58.54 | 5.23 | 6.24 | 69.12 | 3.45 | 2.75 | 68.55 | 2.83 | 6.33 |
| | GPF | 51.82 | 2.68 | 2.48 | 60.17 | 3.16 | 4.23 | 67.65 | 3.62 | 1.79 | 66.28 | 2.42 | 2.50 |
| | GPF+ | 54.50 | 1.15 | 1.44 | 54.67 | 6.29 | 4.35 | 68.52 | 3.49 | 2.03 | 67.20 | 2.12 | **1.07** |
| | Self-pro | 53.03 | 1.41 | **1.41** | 56.83 | 6.67 | 5.07 | 69.18 | 4.77 | 4.23 | 65.58 | 4.09 | 5.43 |
| | FPrompt | 53.72 | 2.26 | 2.08 | 56.00 | 2.97 | 5.73 | 69.05 | **1.08** | 1.33 | 67.75 | 2.37 | 2.80 |
| | ADPrompt | 54.94 | **0.43** | 1.46 | **65.17** | **1.55** | **1.23** | 70.52 | 1.71 | 0.88 | 70.68 | 1.30 | 2.39 |

| Dataset | Prompt Method | ACC (↑) | ΔEO (↓) | ΔSP (↓) |
|---|---|---|---|---|
| Credit | Classifier only | 60.12 | 6.21 | 3.19 |
| | Adversarial learning | 56.00 | 1.88 | 1.34 |
| | GPF | 58.88 | 2.51 | 2.53 |
| | GPF+ | **63.18** | 2.03 | 1.86 |
| | FPrompt | 62.36 | 1.92 | 1.57 |
| | ADPrompt | 62.77 | **1.84** | **1.11** |
| German | Classifier only | 47.50 | 4.87 | 7.46 |
| | Adversarial learning | 52.40 | 2.32 | 2.84 |
| | GPF | 52.19 | 5.37 | 2.41 |
| | GPF+ | **57.43** | 2.12 | 3.94 |
| | FPrompt | 53.28 | 3.04 | 1.99 |
| | ADPrompt | 54.50 | **2.08** | 1.86 |
| Pokec_n | Classifier only | 63.22 | 7.40 | 3.84 |
| | Adversarial learning | 65.72 | 6.44 | 3.34 |
| | GPF | 61.28 | 4.43 | 5.58 |
| | GPF+ | 66.12 | 2.74 | 1.96 |
| | FPrompt | 65.39 | **1.28** | 2.43 |
| | ADPrompt | **66.75** | 1.65 | **1.73** |
| Pokec_z | Classifier only | 65.18 | 11.45 | 4.25 |
| | Adversarial learning | 65.48 | 7.87 | 3.21 |
| | GPF | 64.25 | 5.26 | 3.78 |
| | GPF+ | 65.51 | 4.15 | 2.23 |
| | FPrompt | 63.47 | 2.13 | 3.52 |
| | ADPrompt | **67.43** | **1.02** | **1.85** |

Table 5: Multi-label classification results of different methods.

### E.3 COMPUTATIONAL EFFICIENCY COMPARISON OF PROMPTING METHODS

This section examines the computational efficiency of several prompting methods, including GPF, GPF+, FPrompt, and ADPrompt. Table 6 presents the running time (seconds) and GPU memory usage (GB) under the GraphCL pre-training strategy across two datasets. Although ADPrompt exhibits runtime and memory consumption comparable to other methods, it consistently achieves superior fairness performance, as reported in Table 1, highlighting a favorable balance between computational efficiency and predictive effectiveness.

| Dataset | Prompt Method | Time (s) | GPU (GB) |
|---|---|---|---|
| Credit | GPF | 5.33 | 1.08 |
| | GPF+ | 6.42 | 1.32 |
| | FPrompt | 5.79 | 1.13 |
| | ADPrompt | 5.64 | 1.09 |
| Pokec_n | GPF | 28.42 | 4.17 |
| | GPF+ | 30.18 | 5.04 |
| | FPrompt | 26.32 | 4.78 |
| | ADPrompt | 29.95 | 4.99 |

Table 6: Comparison of different prompt methods under the GraphCL pre-training strategy across datasets.

### E.4 ABLATION STUDY

To demonstrate the efficacy of each module within our proposed method, we conducted a series of ablation experiments, specifically focusing on the AFR and the AMC. Across all four datasets, we consistently utilized the InfoMax pre-training method for these evaluations. As evidenced by the experimental results presented in Figure 5, both constituent parts of our method significantly contribute to enhancing performance on downstream tasks while simultaneously improving fairness.

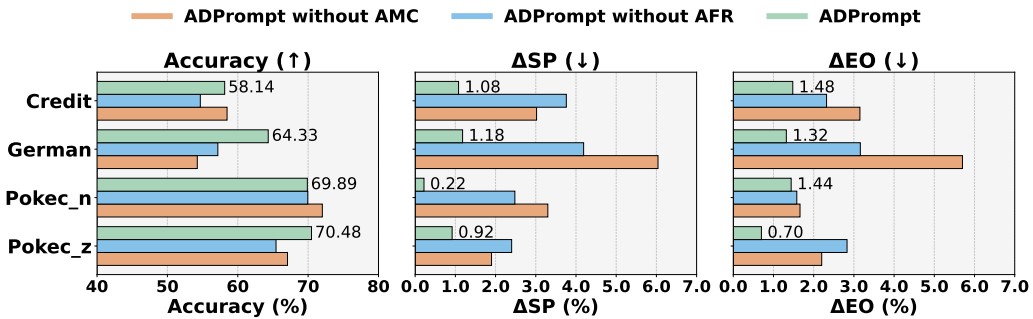

Figure 5: Ablation study of ADPrompt across four datasets under InfoMax pre-training strategy.

### E.5 ABLATION STUDY ON COMPUTATIONAL EFFICIENCY

To evaluate the computational impact of the structure component in AMC, we conducted an ablation study comparing ADPrompt with and without structure prompts across four datasets and two pre-training methods. Table 7 reports the running time (seconds) and GPU memory usage (GB) for each configuration. The experiments are repeated three times and reported the average. The results indicate that including the structure prompts introduces only a modest increase in both computation and memory consumption. These findings confirm that ADPrompt's AMC component is lightweight and practical for large-scale applications.

Table 7: Running time (in seconds) and GPU memory usage (GB) of ADPrompt with and without structure prompts.

| Dataset | Pre-training | Method | Time (s) | GPU (GB) |
|---------|-------------|--------|----------|----------|
| Credit | InfoMax | ADPrompt | 6.19 | 1.08 |
| | | w/o AMC | 5.90 | 0.75 |
| | GraphCL | ADPrompt | 5.64 | 0.19 |
| | | w/o AMC | 4.77 | 0.10 |
| German | InfoMax | ADPrompt | 4.15 | 0.15 |
| | | w/o AMC | 3.77 | 0.08 |
| | GraphCL | ADPrompt | 3.88 | 0.11 |
| | | w/o AMC | 2.99 | 0.08 |
| Pokec_n | InfoMax | ADPrompt | 64.04 | 8.26 |
| | | w/o AMC | 37.46 | 5.76 |
| | GraphCL | ADPrompt | 29.95 | 4.99 |
| | | w/o AMC | 18.27 | 3.60 |
| Pokec_z | InfoMax | ADPrompt | 52.48 | 6.77 |
| | | w/o AMC | 30.17 | 4.31 |
| | GraphCL | ADPrompt | 30.12 | 4.14 |
| | | w/o AMC | 20.46 | 2.82 |

## E.6 PERFORMANCE COMPARISON UNDER DIFFERENT BACKBONE MODELS

We also investigate the performance of ADPrompt with different backbones. Table show the results on the Credit dataset with GraphSage (Hamilton et al., 2017b) and GAT (Veličković et al., 2018a) as backbones pre-trained by InfoMax. From the table, we can observe that our method outperforms three state-of-the-art baselines.

| Backbone | Prompt Method | ACC ($\uparrow$) | $\Delta$EO ($\downarrow$) | $\Delta$SP ($\downarrow$) |
|----------|---------------|---------|---------|---------|
| GraphSage | GPF | 63.29 | 3.05 | 3.97 |
| | GPF+ | 65.55 | 2.30 | 4.13 |
| | FPrompt | 64.63 | 3.13 | 3.40 |
| | ADPrompt | **66.67** | **1.74** | **2.44** |
| GAT | GPF | 57.48 | 2.39 | 2.97 |
| | GPF+ | 61.07 | 3.43 | 1.73 |
| | FPrompt | 56.47 | 2.17 | 4.30 |
| | ADPrompt | **62.67** | **1.74** | **1.70** |

Table 8: Comparison of various methods under different backbone models with InfoMax pre-training strategy on the Credit dataset.

## E.7 PERFORMANCE COMPARISON OF STRUCTURE PROMPT PLACEMENTS

To assess the impact of dynamic message calibration in AMC, we compare ADPrompt with variants that apply the structure prompt only at the first or second layer. As shown in Table 9, across two pre-training strategies (InfoMax and GAE) and four datasets, ADPrompt consistently delivers higher accuracy and smaller fairness gaps, underscoring the benefit of dynamically calibrating structural information across layers rather than restricting it to a single layer.

| Pre-training | Dataset | Method | ACC (↑) | ΔEO (↓) | ΔSP (↓) |
|---|---|---|---|---|---|
| InfoMax | Credit | ADPrompt | **58.14** | **1.08** | **1.48** |
| | | ADPrompt (first layer) | 56.48 | 2.68 | 2.25 |
| | | ADPrompt (second layer) | 55.92 | 2.14 | 2.64 |
| | German | ADPrompt | **64.33** | **1.18** | **1.32** |
| | | ADPrompt (first layer) | 62.89 | 2.53 | 3.27 |
| | | ADPrompt (second layer) | 59.24 | 2.66 | 5.75 |
| | Pokec_n | ADPrompt | **69.89** | **0.22** | **1.44** |
| | | ADPrompt (first layer) | 67.86 | 1.06 | 1.16 |
| | | ADPrompt (second layer) | 62.72 | 5.11 | 3.23 |
| | Pokec_z | ADPrompt | **70.48** | **0.92** | **0.70** |
| | | ADPrompt (first layer) | 65.28 | 4.38 | 2.15 |
| | | ADPrompt (second layer) | 67.74 | 6.56 | 5.59 |
| GAE | Credit | ADPrompt | **64.86** | **1.37** | **1.28** |
| | | ADPrompt (first layer) | 58.48 | 5.69 | 5.13 |
| | | ADPrompt (second layer) | 54.68 | 1.56 | 1.43 |
| | German | ADPrompt | **62.17** | **2.22** | **2.32** |
| | | ADPrompt (first layer) | 59.25 | 5.70 | 3.78 |
| | | ADPrompt (second layer) | 54.68 | 5.48 | 2.59 |
| | Pokec_n | ADPrompt | **73.89** | **0.22** | **1.44** |
| | | ADPrompt (first layer) | 58.48 | 3.02 | 3.15 |
| | | ADPrompt (second layer) | 54.68 | 3.76 | 2.32 |
| | Pokec_z | ADPrompt | **75.09** | **1.25** | **0.84** |
| | | ADPrompt (first layer) | 70.32 | 2.28 | 1.96 |
| | | ADPrompt (second layer) | 73.56 | 3.42 | 1.04 |

Table 9: Performance comparison with different structure prompt placements.

# F USE OF LARGE LANGUAGE MODELS

Large language models (LLMs) were used solely for polishing the writing of this paper. No other uses of LLMs were involved.

