# OpenReview forum: "Adaptive Dual Prompting: Hierarchical Debiasing for Fairness-aware Graph Neural Networks"
_ICLR.cc/2026/Conference — ICLR 2026 Conference Withdrawn Submission_

### Official Review · Reviewer_w2LT · 2025-10-24

**Soundness:** 2
**Presentation:** 2
**Contribution:** 3
**Rating:** 4
**Confidence:** 4

**Summary:**

The paper introduces ADPrompt, a fairness-aware prompting framework for adapting pre-trained GNNs to downstream node classification while improving group fairness. It comprises two modules: Adaptive Feature Rectification (AFR), which gates feature dimensions via learnable attribute prompts to suppress sensitive information, and Adaptive Message Calibration (AMC), which injects edge-specific structure prompts at each layer to calibrate message passing. Experiments on four datasets and four pre-training strategies show higher accuracy with lower $\Delta\mathrm{EO}/\Delta\mathrm{SP}$ than seven baselines.

**Strengths:**

+ The modular method is compatible with frozen backbones. AFR and AMC are lightweight prompts on features and messages, easy to add to existing GNNs.
+ The theoretical results are tied to design. The $\Delta\mathrm{GSP}$ upper bound links AFR to reduced initial bias and AMC to damped propagation amplification.
+ Experiments are comprehensive. Four datasets $\times$ four pre-training schemes with seven baselines demonstrate the method's effectiveness.

**Weaknesses:**

- The work is restricted to binary $y$ and a single binary $s$. How about multi-class or multi-attribute evaluation?
- AMC learns an edge-specific prompt $e^{(l-1)}_{ij}$ at each layer, implying $\mathcal{O}(|E|\cdot d \cdot L)$ memory/compute overhead. The paper does not report runtime/memory comparisons with baselines.
- The fairness bound relies on Lipschitz assumptions and multiplicative factors $\tilde{\gamma}^{(l)}, \tilde{\epsilon}^{(l)}$, but the paper provides no estimators or empirical diagnostics for these constants. It is unclear how the training losses control the bound.
- Individual fairness is not assessed, though structural edits may alter local similarities.
- Potential error amplification from mislabeled $y/s$ is not analyzed.

**Questions:**

Please refer to the above weaknesses.

---

> ### Author Response · Authors · 2025-11-20
> **Author Response to Reviewer w2LT (1/2)**
>
> We sincerely appreciate your efforts to review our paper and provide insightful suggestions. We hope our following point-by-point clarifications can address your concerns.
>
> ---
>
> **W1:** The work is restricted to binary y and a single binary s. How about multi-class or multi-attribute evaluation?
>
> **R1:** Thank you for your insightful suggestion. We have added experiments on **multi-label classification** to demonstrate the method's versatility. Please see **Appendix E.2** and the new **Table 5** in the revised PDF. ADPrompt consistently ranks top in accuracy while maintaining low fairness disparities, demonstrating its generalizability beyond binary attributes.
>
> Table 5: Performance on Multi-label Classification Tasks
>
> | Dataset | Prompt Method | ACC ($\uparrow$) | $\Delta$EO ($\downarrow$) | $\Delta$SP ($\downarrow$) |
> | :--- | :--- | :---: | :---: | :---: |
> | Credit | Classifier only | 60.12 | 6.21 | 3.19 |
> | | Adversarial learning | 56.00 | 1.88 | 1.34 |
> | | GPF | 58.88 | 2.51 | 2.53 |
> | | GPF+ | 63.18 | 2.03 | 1.86 |
> | | FPrompt | 62.36 | 1.92 | 1.57 |
> | | ADPrompt | 62.77 | 1.84 | 1.11 |
> | German | Classifier only | 47.50 | 4.87 | 7.46 |
> | | Adversarial learning | 52.40 | 2.32 | 2.84 |
> | | GPF | 52.19 | 5.37 | 2.41 |
> | | GPF+ | 57.43 | 2.12 | 3.94 |
> | | FPrompt | 53.28 | 3.04 | 1.99 |
> | | ADPrompt | 54.50 | 2.08 | 1.86 |
> | Pokec-n | Classifier only | 63.22 | 7.40 | 3.84 |
> | | Adversarial learning | 65.72 | 6.44 | 3.34 |
> | | GPF | 61.28 | 4.43 | 5.58 |
> | | GPF+ | 66.12 | 2.74 | 1.96 |
> | | FPrompt | 65.39 | 1.28 | 2.43 |
> | | ADPrompt | 66.75 | 1.65 | 1.73 |
> | Pokec-z | Classifier only | 65.18 | 11.45 | 4.25 |
> | | Adversarial learning | 65.48 | 7.87 | 3.21 |
> | | GPF | 64.25 | 5.26 | 3.78 |
> | | GPF+ | 65.51 | 4.15 | 2.23 |
> | | FPrompt | 63.47 | 2.13 | 3.52 |
> | | ADPrompt | 67.43 | 1.02 | 1.85 |
>
> ---
>
> **W2:** AMC learns an edge-specific prompt $e^{(l-1)}_{ij}$ at each layer, implying $\mathcal{O}(|E|\cdot d \cdot L)$ memory/compute overhead. The paper does not report runtime/memory comparisons with baselines.
>
> **R2:** We appreciate you pointing out the need for a more detailed efficiency analysis. We have added a comprehensive evaluation in the Appendix to address the memory and compute concerns regarding the edge-specific prompt $e^{(l-1)}_{ij}$.
>
> 1.  **Ablation on AMC Cost:** In **Appendix E.5 (Table 7)**, we analyze the cost of the AMC module by running time and GPU usage. The results indicate that structure prompts introduce only a modest increase in computation (e.g., Credit dataset time increases from 4.77s to 5.64s), confirming AMC is lightweight.
> 2.  **Comparison with Baselines:** In **Appendix E.3 (Table 6)**, we compare the running time and GPU usage of ADPrompt with GPF, GPF+, and FPrompt. ADPrompt shows comparable efficiency while achieving superior fairness.
>
> Table 7: Running time and GPU usage of ADPrompt with and without structure prompts
>
> | Dataset   | Pre-training | Method      | Time (s) | GPU (GB) |
> |-----------|--------------|------------|----------|----------|
> | Credit    | InfoMax      | ADPrompt   | 6.19     | 1.08     |
> |           |              | w/o AMC    | 5.90     | 0.75     |
> |           | GraphCL      | ADPrompt   | 5.64     | 0.19     |
> |           |              | w/o AMC    | 4.77     | 0.10     |
> | German    | InfoMax      | ADPrompt   | 4.15     | 0.15     |
> |           |              | w/o AMC    | 3.77     | 0.08     |
> |           | GraphCL      | ADPrompt   | 3.88     | 0.11     |
> |           |              | w/o AMC    | 2.99     | 0.08     |
> | Pokec-n   | InfoMax      | ADPrompt   | 64.04    | 8.26     |
> |           |              | w/o AMC    | 37.46    | 5.76     |
> |           | GraphCL      | ADPrompt   | 29.95    | 4.99     |
> |           |              | w/o AMC    | 18.27    | 3.60     |
> | Pokec-z   | InfoMax      | ADPrompt   | 52.48    | 6.77     |
> |           |              | w/o AMC    | 30.17    | 4.31     |
> |           | GraphCL      | ADPrompt   | 30.12    | 4.14     |
> |           |              | w/o AMC    | 20.46    | 2.82     |
>
> Table 6: Efficiency Comparison with Baseline Prompting Methods
>
> | Dataset | Prompt Method | Time (s) | GPU (GB) |
> | :--- | :--- | :---: | :---: |
> | Credit | GPF | 5.33 | 1.08 |
> | | GPF+ | 6.42 | 1.32 |
> | | FPrompt | 5.79 | 1.13 |
> | | ADPrompt | 5.64 | 1.09 |
> | Pokec-n | GPF | 28.42 | 4.17 |
> | | GPF+ | 30.18 | 5.04 |
> | | FPrompt | 26.32 | 4.78 |
> | | ADPrompt | 29.95 | 4.99 |

---

> ### Author Response · Authors · 2025-11-20
> **Author Response to Reviewer w2LT (2/2)**
>
> **W3:** The fairness bound relies on Lipschitz assumptions and multiplicative factors $\pi^{(l)}$, $\epsilon^{(l)}$, but the paper provides no estimators or empirical diagnostics for these constants. It is unclear how the training losses control the bound.
>
> **R3:** Thanks for raising this point. While the exact tightness of the bound depends on fixed Lipschitz and structural constants, Theorem 1 mainly acts as an optimization guide. It identifies which trainable parts of the model are directly influenced by the loss terms, showing how training gradually tightens the theoretical upper bound.
>
> 1. **Initial Bias Control (AFR):**  The adversarial loss encourages the rectified feature \\( \tilde{X} \\) to be independent of \\( S \\). This reduces the initial feature bias \\( \Delta_{\text{GSP}}(\tilde{X}) \\), lowering the starting bias in the input.
>
> 2. **Propagation Bias Control (AMC):**  The task loss optimizes the AMC module, reducing the structural residuals \\( \tilde{\epsilon}^{(l)} \\) and the propagation factors \\( \tilde{\gamma}^{(l)} \\) at each layer.
>
> By minimizing these two controllable sources of bias, the training process keeps the final upper bound on \\( \Delta_{\text{GSP}}(\tilde{h}^{(L)}) \\) effectively constrained.
>
>
>
> ---
>
> **W4:** Individual fairness is not assessed, though structural edits may alter local similarities.
>
> **R4:** We appreciate this suggestion. But we clarify in **Section 3.2** that the scope of this work is **Group Fairness** (group Statistical Parity and Equal Opportunity), which is the standard focus in the graph fairness literature. Individual fairness requires different metrics and objectives, which are outside the scope of this paper.
>
> ---
>
> **W5:** Potential error amplification from mislabeled y/s is not analyzed.
>
> **R5:** Thank you for raising the concern. In our experiments, we follow the standard evaluation protocol using ground-truth labels provided by the datasets. Dealing with noisy or mislabeled data falls under the domain of Robustness against Label Noise. Our study focus on improving group fairness. While this is a valid concern in real-world deployments, it is a different research topic.

---

### Official Review · Reviewer_w87U · 2025-10-30

**Soundness:** 3
**Presentation:** 2
**Contribution:** 2
**Rating:** 4
**Confidence:** 5

**Summary:**

The paper proposes Adaptive Dual Prompting (ADPrompt), a fairness-aware graph prompting framework for adapting pre-trained GNN backbones to downstream node classification while improving group fairness. It introduces two complementary prompt modules: (i) Adaptive Feature Rectification (AFR), a self-gated attribute prompt that suppresses sensitive information at the input; and (ii) Adaptive Message Calibration (AMC), edge- and layer-specific structure prompts to softly calibrate message passing. A min–max objective combines supervised training with an adversary predicting sensitive attributes from the prompted representations. Theoretical analysis bounds group disparity across layers, and experiments on four datasets demonstrate the effectiveness of the proposed method compared to existing baselines.

**Strengths:**

1. The idea is straightforward and easy to follow.

2. The theoretical analysis is provided.

3. The experimental results across four datasets demonstrate the empirical effectiveness of the proposed method.

**Weaknesses:**

1. The writing is unclear and can be largely improved. Specifically,  (i) in Section 1 (Introduction), the authors fail to claim why we need this proposed method instead of existing fairness graph prompt methods, such as [1]. The challenges mentioned in this section are merely some well-known fairness issues of GNN, making the reasons to design the proposed method unclear; (ii) The contributions mentioned in Section 1 should also be largely rewritten. The first two points are literally the same thing.

2. In Section 4 (Methodology), while some limitations of existing methods are mentioned, these points are not convincing. For example, the author claims that FPrompt [1] may disrupt critical topological information. However, it is unclear how and why it will disrupt critical topological information and lead to a performance drop. I can only find the underlying reason when I read Section 4.2. The authors are suggested to rewrite this section to make it clearer to readers. In addition, if possible, please add some preliminary empirical results to validate the claims. And I think the above clarification should also be mentioned in Section 1.

3. While theoretical analysis is provided, the statement in Section 5.2 is not clear enough to conclude Theorem 1. I understand that this subsection aims to give proofs and propose Theorem 1 to validate the effectiveness of the proposed method. However, a detailed proof for Theorem 1 should be included. Otherwise, it is unclear how Theorem 1 comes from and why Eq. (16) can lead to Theorem 1.

4. Computational overhead. AMC learns edge- and layer-specific structure prompts, which may impose significant memory and time costs on dense graphs or deep GNN backbones. The paper omits complexity analysis and runtime/memory profiling relative to baselines.

5. It seems that the framework presumes a binary sensitive attribute is known for all nodes. The AFR and adversary rely on this supervision. Real-world graphs often have missing or multi-valued sensitive labels. More discussion and analysis on this scenario can benefit this paper.

6. Current ablation studies focus on removing AFR or AMC. However, it is also important to investigate the sensitivity of the hyperparameter,s such as $\lambda$, and the transferability of the proposed method. And the impact of layer numbers of GNNs is also important to investigate the effectiveness of the layer-specific techniques.

7. Too many critical analyses are put into Appendix. The authors are suggested to reorganize this paper such that some important analysis can be in the main text.

[1] Fairness-aware prompt tuning for graph neural networks. WWW 2025.

**Questions:**

1. Scalability analysis. Can the authors provide runtime/memory comparisons between ADPrompt and GPF/FPrompt on large graphs? Are there ways to sparsify AMC prompts (e.g., top‑k neighbors or low-rank factorization)?

2. Transferability of prompts. If prompts are learned on one dataset or under one pre‑training method, can they be transferred to another domain or backbone without retraining? Preliminary results would be interesting.

---

> ### Author Response · Authors · 2025-11-20
> **Author Response to Reviewer w87U (1/3)**
>
> We thank the reviewer for the detailed feedback. We hope our following point-by-point clarifications can address your concerns.
>
> ----
>
> **W1:** (i) in Section 1 (Introduction), the authors fail to claim why we need this proposed method instead of existing fairness graph prompt methods, such as [1]. The challenges mentioned in this section are merely some well-known fairness issues of GNN, making the reasons to design the proposed method unclear; (ii) The contributions mentioned in Section 1 should also be largely rewritten. The first two points are literally the same thing.
>
> **R1:**
> (i) **Necessity of ADPrompt:** We have clarified the challenges in **Section 1 (Lines 84-102)**. The only existing fairness prompting work, FPrompt, primarily focuses on attribute bias and uses hard interventions (edge editing). Existing methods largely neglect **structure bias** in the message-passing flow. ADPrompt is the first to address both attribute and structure through the whole message flow via adaptive feature rectification and message calibration.  The detailed motivation is discussed in introduction.
> (ii) **Contributions:** We have rewritten the contributions in **Introduction (Lines 120-122)** and highlighted the part in red in our revised PDF. Our contribution mainly focuses on framework design, theoretical analysis, and experimental evaluation.
>
> ---
>
> **W2:** In Section 4 (Methodology), while some limitations of existing methods are mentioned, these points are not convincing. For example, the author claims that FPrompt [1] may disrupt critical topological information. However, it is unclear how and why it will disrupt critical topological information and lead to a performance drop.
>
> **R2:** We have revised **Section 4 and 4.2 (Lines 244-247)** to clarify this. FPrompt and some related methods often employ "hard interventions" (deleting or adding edges) to mitigate bias, which rigidly alters the graph topology and may destroy semantic relationships (e.g., removing a "follow" link in a social network). In contrast, ADPrompt uses Adaptive Message Calibration (AMC), which assigns soft, learnable weights to messages, preserving the original graph connectivity while dynamically correcting biased information flow.
>
> ---
>
>
> **W3:** While theoretical analysis is provided, the statement in Section 5.2 is not clear enough to conclude Theorem 1. A detailed proof for Theorem 1 should be included.
>
> **R3:** We have added the complete, step-by-step proof for Theorem 1 in **Section 5.2 (Lines 341-408)** of the revised paper. This includes the derivation from the initial feature bias (Eq. 15) and the layer-wise propagation bound (Eq. 18), formally establishing the theoretical foundation of ADPrompt.
>
> ---
>
> **W4:** Computational overhead. AMC learns edge- and layer-specific structure prompts, which may impose significant memory and time costs on dense graphs or deep GNN backbones. The paper omits complexity analysis and runtime/memory profiling relative to baselines.
>
> **R4:** We have added a comprehensive efficiency analysis in the Appendix.
>
> 1. **Ablation on AMC Cost:** In **Appendix E.5 (Table 7)**, we analyze the cost of the AMC module by the running time and GPU usage. The results indicate that structure prompts introduce only a modest increase in computation (e.g., Credit dataset time increases from 4.77s to 5.64s), confirming AMC is lightweight.
> 2. **Comparison with Baselines:** In **Appendix E.3 (Table 6)**, we compare the running time and GPU usage of ADPrompt with GPF, GPF+ and FPrompt. ADPrompt shows comparable efficiency while achieving superior fairness.
>
> Here is the result: Running time and GPU usage of ADPrompt with and without structure prompts (seconds / GB)
>
> | Dataset | Pre-training | Method    | Time (s) | GPU (GB) |
> |---------|--------------|-----------|----------|----------|
> | Credit  | InfoMax      | ADPrompt  | 6.19     | 1.08     |
> | Credit  | InfoMax      | w/o AMC   | 5.90     | 0.75     |
> | Credit  | GraphCL      | ADPrompt  | 5.64     | 0.19     |
> | Credit  | GraphCL      | w/o AMC   | 4.77     | 0.75     |
> | German  | InfoMax      | ADPrompt  | 4.15     | 0.15     |
> | German  | InfoMax      | w/o AMC   | 3.77     | 0.08     |
> | German  | GraphCL      | ADPrompt  | 3.88     | 0.11     |
> | German  | GraphCL      | w/o AMC   | 2.99     | 0.08     |
>
> ---
>
> Here is the result: Comparison with baseline prompting methods (Credit and Pokec-n datasets):
>
> | Dataset  | Prompt Method | Time (s) | GPU (GB) |
> |----------|---------------|----------|----------|
> | Credit   | GPF           | 5.33     | 1.08     |
> | Credit   | GPF+          | 6.42     | 1.32     |
> | Credit   | FPrompt       | 5.79     | 1.13     |
> | Credit   | ADPrompt      | 5.64     | 1.09     |
> | Pokec-n  | GPF           | 28.42    | 4.17     |
> | Pokec-n  | GPF+          | 30.18    | 5.04     |
> | Pokec-n  | FPrompt       | 26.32    | 4.78     |
> | Pokec-n  | ADPrompt      | 29.95    | 4.99     |

---

> ### Author Response · Authors · 2025-11-20
> **Author Response to Reviewer w87U (2/3)**
>
> **W5:** It seems that the framework presumes a binary sensitive attribute is known for all nodes. The AFR and adversary rely on this supervision. Real-world graphs often have missing or multi-valued sensitive labels. More discussion and analysis on this scenario can benefit this paper.
>
> **R5:** To address this, we extended our experiment to **multi-label classification tasks**. The results are reported in **Appendix E.2 and Table 5**. ADPrompt consistently ranks top in accuracy while maintaining low fairness disparities, demonstrating its generalizability beyond binary attributes.
>
> Results of multi-label classification tasks:
>
> | Dataset | Prompt Method | ACC (↑) | ΔEO (↓) | ΔSP (↓) |
> | :--- | :--- | :--- | :--- | :--- |
> | Credit | Classifier only | 60.12 | 6.21 | 3.19 |
> | | Adversarial learning | 56.00 | 1.88 | 1.34 |
> | | GPF | 58.88 | 2.51 | 2.53 |
> | | GPF+ | 63.18 | 2.03 | 1.86 |
> | | FPrompt | 62.36 | 1.92 | 1.57 |
> | | ADPrompt | 62.77 | 1.84 | 1.11 |
> | German | Classifier only | 47.50 | 4.87 | 7.46 |
> | | Adversarial learning | 52.40 | 2.32 | 2.84 |
> | | GPF | 52.19 | 5.37 | 2.41 |
> | | GPF+ | 57.43 | 2.12 | 3.94 |
> | | FPrompt | 53.28 | 3.04 | 1.99 |
> | | ADPrompt | 54.50 | 2.08 | 1.86 |
> | Pokec-n | Classifier only | 63.22 | 7.40 | 3.84 |
> | | Adversarial learning | 65.72 | 6.44 | 3.34 |
> | | GPF | 61.28 | 4.43 | 5.58 |
> | | GPF+ | 66.12 | 2.74 | 1.96 |
> | | FPrompt | 65.39 | 1.28 | 2.43 |
> | | ADPrompt | 66.75 | 1.65 | 1.73 |
> | Pokec-z | Classifier only | 65.18 | 11.45 | 4.25 |
> | | Adversarial learning | 65.48 | 7.87 | 3.21 |
> | | GPF | 64.25 | 5.26 | 3.78 |
> | | GPF+ | 65.51 | 4.15 | 2.23 |
> | | FPrompt | 63.47 | 2.13 | 3.52 |
> | | ADPrompt | 67.43 | 1.02 | 1.85 |
>
>
> ---
>
> **W6:** Current ablation studies focus on removing AFR or AMC. However, it is also important to investigate the sensitivity of the hyperparameter,s such as $\lambda$, and the transferability of the proposed method. And the impact of layer numbers of GNNs is also important to investigate the effectiveness of the layer-specific techniques.
>
> **R6:**
> 1. **Hyperparameter λ:** We analyzed the sensitivity of λ in **Figure 4 (Section 6.3 Impact of the Balancing Parameter)**. The results show a stable performance trade-off in the range of λ ∈ [5, 7].
> 2. **Layer Numbers:** We investigated the impact of applying prompts at different layers in **Appendix E.7 (Table 9)**. The results confirm that applying AMC across all layers consistently outperforms applying it only to the first or second layer.
>
> Results of performance comparison of structure prompt placements:
>
> | Pre-training | Dataset | Method | ACC (↑) | $\Delta$EO (↓) | $\Delta$SP (↓) |
> | :--- | :--- | :--- | :--- | :--- | :--- |
> | InfoMax | Credit | ADPrompt | 58.14 | 1.08 | 1.48 |
> | | | ADPrompt (first layer) | 56.48 | 2.68 | 2.25 |
> | | | ADPrompt (second layer) | 55.92 | 2.14 | 2.64 |
> | | German | ADPrompt | 64.33 | 1.18 | 1.32 |
> | | | ADPrompt (first layer) | 62.89 | 2.53 | 3.27 |
> | | | ADPrompt (second layer) | 59.24 | 2.66 | 5.75 |
> | | Pokec\_n | ADPrompt | 69.89 | 0.22 | 1.44 |
> | | | ADPrompt (first layer) | 67.86 | 1.06 | 1.16 |
> | | | ADPrompt (second layer) | 62.72 | 5.11 | 3.23 |
> | | Pokec\_z | ADPrompt | 70.48 | 0.92 | 0.70 |
> | | | ADPrompt (first layer) | 65.28 | 4.38 | 2.15 |
> | | | ADPrompt (second layer) | 67.74 | 6.56 | 5.59 |
> | GAE | Credit | ADPrompt | 64.86 | 1.37 | 1.28 |
> | | | ADPrompt (first layer) | 58.48 | 5.69 | 5.13 |
> | | | ADPrompt (second layer) | 54.68 | 1.56 | 1.43 |
> | | German | ADPrompt | 62.17 | 2.22 | 2.32 |
> | | | ADPrompt (first layer) | 59.25 | 5.70 | 3.78 |
> | | | ADPrompt (second layer) | 54.68 | 5.48 | 2.59 |
> | | Pokec\_n | ADPrompt | 73.89 | 0.22 | 1.44 |
> | | | ADPrompt (first layer) | 58.48 | 3.02 | 3.15 |
> | | | ADPrompt (second layer) | 54.68 | 3.76 | 2.32 |
> | | Pokec\_z | ADPrompt | 75.09 | 1.25 | 0.84 |
> | | | ADPrompt (first layer) | 70.32 | 2.28 | 1.96 |
> | | | ADPrompt (second layer) | 73.56 | 3.42 | 1.04 |

---

> ### Author Response · Authors · 2025-11-20
> **Author Response to Reviewer w87U (3/3)**
>
> **W7**: Too many critical analyses are put into Appendix. The authors are suggested to reorganize this paper such that some important analysis can be in the main text.
>
> **R7**: Thanks for pointing it out. In the revised version, we have reorganized the paper to move the core Theoretical Analysis (Section 5) and key efficiency discussions into the main text to improve readability.
>
> ---
>
> **Q1**: Scalability analysis. Can the authors provide runtime/memory comparisons between ADPrompt and GPF/FPrompt on large graphs? Are there ways to sparsify AMC prompts (e.g., top‑k neighbors or low-rank factorization)?
>
> **A1**: Thanks for bringing this up. Please refer to **Tables 2 and 6** for the runtime/memory scalability analysis. Regarding sparsity, we agree that sparsifying AMC prompts is a promising direction for future work to further reduce overhead on extremely dense graphs.
>
> ---
>
> **Q2**: Transferability of prompts. If prompts are learned on one dataset or under one pre‑training method, can they be transferred to another domain or backbone without retraining? Preliminary results would be interesting.
>
> **A2**: We appreciate this interesting suggestion. While prompts are typically dataset-specific in the graph domain, investigating cross-domain transferability (e.g., training prompts on one social network and applying them to another) is a valuable direction for future research.

---

### Official Review · Reviewer_CtSJ · 2025-10-31

**Soundness:** 2
**Presentation:** 2
**Contribution:** 1
**Rating:** 2
**Confidence:** 3

**Summary:**

This paper presents ADPrompt, a fairness-aware prompting framework for adapting pre-trained GNNs to downstream tasks. The core idea involves two adaptive prompting modules: an Adaptive Feature Rectification (AFR) module that purifies node attributes at the input layer to suppress sensitive information, and an Adaptive Message Calibration (AMC) module that dynamically adjusts the message-passing between nodes at each GNN layer to mitigate structural bias. By jointly optimizing these lightweight prompts with a combination of supervised and adversarial losses, the method aims to enhance fairness while maintaining task utility, without updating the frozen pre-trained GNN parameters. Extensive experiments on four datasets under various pre-training strategies demonstrate its effectiveness.

**Strengths:**

1. This paper is well-written and easy to understand

2. This paper grounds its proposed method, ADPrompt, in a robust theoretical framework.

**Weaknesses:**

1. The paper's primary motivation—using graph prompting for fairness—rests on the assumption that pre-trained GNNs are a valuable and widely adopted resource that should be efficiently adapted. However, this premise is not thoroughly debated. In contrast to large language models or vision transformers, GNNs are often task-specific and can be trained from scratch relatively quickly and efficiently. The claimed benefit of prompting—parameter efficiency by freezing the backbone—is less compelling when the backbone itself (a GNN) is not an exceptionally large or general-purpose model. The paper would be stronger if it provided a more convincing justification for why prompting is the right paradigm for this problem, compared to simply building a fairness-aware objective into an end-to-end GNN training process, which is common in graph fairness literature.

2. The baselines are mostly graph prompting methods, lacking dedicated, state-of-the-art graph debiasing methods that do not rely on pre-training or prompting (e.g., Edits [1], FairVGNN [2]).

[1] EDITS: Modeling and Mitigating Data Bias for Graph Neural Networks

[2] Improving Fairness in Graph Neural Networks via Mitigating Sensitive Attribute Leakage

**Questions:**

Please see weaknesses

---

> ### Author Response · Authors · 2025-11-20
> **Author Response to Reviewer CtSJ**
>
> We sincerely appreciate your efforts and provide valuable suggestions. We hope our following point-by-point clarifications can address your concerns.
>
> ---
>
>
> **W1:** The paper's primary motivation... GNNs are often task-specific and can be trained from scratch relatively quickly and efficiently... The paper would be stronger if it provided a more convincing justification for why prompting is the right paradigm...
>
> **R1:** We thank the reviewer for the valuable feedback. We believe there may be some slight misunderstanding, and we would like to clarify our motivation regarding two key aspects: computational cost and label scarcity.
>
> 1. **Computational Cost on Large-Scale Graphs:** While small GNNs can be trained efficiently, real-world graphs have grown tremendously in scale, with massive numbers of nodes and edges. For example, OGB-papers100M (111M nodes, 1.6B edges) [1], Twitter-2010 (41M nodes, 1.47B edges) [2], and Friendster (65M nodes, 1.8B edges) [3] require massive industrial-grade computing resources to train. Retraining a GNN from scratch for every new downstream task is computationally prohibitive. Prompting allows us to reuse one expensive pre-trained model for many tasks.
>
> 2. **Label Scarcity:** The motivation for using graph prompting stems from the scarcity of labeled data. End-to-end training of GNNs requires substantial labeled data, which is often unavailable in practice [4]. To address this, we typically pre-train GNNs on large unlabeled datasets and use lightweight prompts to adapt them to specific downstream tasks. Our prompting approach freezes the pre-trained backbone and introduces only a small set of learnable prompts, allowing for high parameter efficiency. A detailed discussion can be found in the Introduction (Lines 39–75).
>
> References:
>
> [1] Hu et al., "Open Graph Benchmark: Datasets for Machine Learning on Graphs," NeurIPS 2020.
> [2] Kwak et al., "What is Twitter, a Social Network or a News Media?" WWW 2010.
> [3] Yang & Leskovec, "Defining and Evaluating Network Communities based on Ground-Truth," ICDM 2012.
> [4] Fang, T., Zhang, Y., Yang, Y., Wang, C., & Chen, L., "Universal Prompt Tuning for Graph Neural Networks," NeurIPS 2023.
>
> ---
>
> **W2:** The baselines are mostly graph prompting methods, lacking dedicated, state-of-the-art graph debiasing methods (e.g., Edits [1], FairVGNN [2]).
>
> **R2:** Thank you for the suggestion. The reason we primarily compared against prompting methods is **the difference in problem settings**:
>
> (1) Graph Debiasing Methods (e.g., FairVGNN, Edits): These typically assume an **end-to-end supervised setting** where the GNN backbone is trainable and abundant labels are available.
> (2) Our Setting (Pre-training + Prompting): We operate under a setting where the **backbone is frozen (pre-trained on unlabeled data) and labels are extremely scarce** (e.g., 10-shot or 50-shot) during adaptation.
>
> Directly comparing a fully supervised end-to-end method against prompting method would be unfair due to the disparity in trainable parameters and label usage. We believe comparing against fairness-aware prompting and general prompting methods is the most rigorous evaluation for our specific contribution.

---

> > ### Comment · Reviewer_CtSJ · 2025-11-28
> > **Reviewer Response**
> >
> > Thanks for the author's response. However, I disagree with R2 (2). The EDITS framework is a model-agnostic, pre-processing approach, which operates **independently of any specific GNN or downstream task labels**. The method is not an end-to-end supervised setting.
> >
> > It remains unclear to me what advantages this work holds over fairness models that rely on data debiasing, such as EDITS.

---

### Official Review · Reviewer_9XoH · 2025-11-03

**Soundness:** 3
**Presentation:** 3
**Contribution:** 3
**Rating:** 6
**Confidence:** 3

**Summary:**

This paper studies a fairness-aware graph prompting method called ADPrompt that integrates Adaptive Feature Rectification and Adaptive Message Calibration to mitigate biases in both node attributes and graph structure. This method reduces group prejudice in GNNs and also improves adaptability for downstream tasks. The authors also give empirical results on multiple datasets that ADPrompt outperforms some baselines.

**Strengths:**

Theorem 1 gives a fairness guarantee of ADPrompt, which shows that ADPrompt reduces initial feature bias and suppresses
bias propagation, providing a tighter upper bound on Δ_GSP than a standard GNN. The authors also provide empirical result to justify their theoretical findings. This results are very interesting.

**Weaknesses:**

Theorem 1 only shows a relationship of less or equal than, but does not really tell how much higher the inequality is.

**Questions:**

1. Can you explain how tight Eq. 12 and 16 are? What is the best possible inequality (i.e., the limit) of Eq. 17?

2. Where are the complete proofs of the results in Section 5.2? I cannot check the whole details in the current version.

3. The empirical results show that their proposed method achieves the best or highly competitive performance across various pre-training strategies (Table 1). My question is, does a method that can better suppress the bias always show better performance? Can we analyze it quantitatively?

4. Small issue: after Theorem 1, the explanation "This formally shows that ADPrompt reduces initial feature bias and suppresses bias propagation, providing a tighter upper bound on ΔGSP than a standard GNN." should be not part of the theorem.

---

> ### Author Response · Authors · 2025-11-20
> **Author Response to Reviewer 9XoH**
>
> We sincerely appreciate your efforts to review our paper and provide valuable suggestions. We hope our following point-by-point clarifications can address your concerns.
>
> ---
>
> **W1:** Theorem 1 only shows a relationship of less or equal than, but does not really tell how much higher the inequality is.
>
> **R:** We acknowledge that quantifying the tightness of the inequality is challenging, as it inherently depends on the intrinsic data distribution and the specific Lipschitz constants of the network layers. We revise our PDF revision and highlight the polished **Section 5.2(Fairness Bound of ADPrompt)** in blue, which can provide more detailed and rigorous theoretical demonstration
> . The primary value of Theorem 1 lies in its decomposition of bias sources: the initial feature bias and the bias amplified during propagation. Through AFR and AMC, ADPrompt effectively tightens the overall upper bound on $\Delta_{GSP}$. Also, the empirical success of ADPrompt, which consistently achieving lower $\Delta$EO and $\Delta$SP across all experiments, strongly validates that our method can minimize this bound  and reduce actual group disparity.
>
> ---
>
> **Q1:** Can you explain how tight Eq.12 and Eq.16 are? What is the best possible inequality (i.e., the limit) of Eq.17?
>
> **R1:** Thank you for this insightful question regarding the theoretical bounds. In the revised manuscript, the original Eq.12, 16, and 17 correspond to **Eq.15, 18, and 12**, respectively, and their complete derivations are now provided in **Section 5.2 (Fairness Bound of ADPrompt)**.
>
> For Eq.15 (initial feature bias), the tightness of the inequality $\Delta_{GSP}(\tilde{X}) \le \Delta_{GSP}(X)$ depends on the effectiveness of the Adaptive Feature Rectification (AFR) module. In the theoretical best-case scenario, $\Delta_{GSP}(\tilde{X})$ approaches $0$ (equivalently, $I(\tilde{X}; S) \to 0$), meaning that the AFR module can almost completely remove feature bias from the input.
>
> For Eq.18 (layer-wise propagation), the tightest bound appears when the Adaptive Message Calibration (AMC) module fully suppresses bias amplification, i.e.,  $\tilde{\gamma}^{(l)} \to 0$ and $\tilde{\epsilon}^{(l)} \to 0$.
>
> Eq.12 (final theorem) integrates both effects. Its optimal theoretical limit is  $\Delta_{GSP}(\tilde{h}^{(L)}) \to 0$,  achievable only if AFR minimizes the initial feature bias and AMC blocks bias propagation across all layers.
>
> ---
>
> **Q2:** Where are the complete proofs of the results in Section 5.2? I cannot check the whole details in the current version.
>
> **R2:** We apologize for the omission in the initial submission due to page limits. The revised manuscript now includes the **complete and rigorous proofs** in **Section 5.2 (Lines 338–403)**.
>
> Theorem 1 is established in two steps:
> (1) Proof 1 uses Pinsker’s inequality and mutual information analysis to show that AFR reduces the initial feature bias.
> (2) Proof 2 applies Lipschitz continuity and recursive unrolling to demonstrate how AMC progressively tightens the amplification factor $\tilde{\gamma}$ and residual $\tilde{\epsilon}$, producing the final form of Eq.12. The derivation is performed in Eq.18 and 19.
>
> ---
>
> **Q3:** The empirical results show that the proposed method achieves the best performance. Does stronger debiasing always imply better performance? Can this be analyzed quantitatively?
>
> **R3:** To answer this quantitatively, we conducted a sensitivity analysis of the balancing parameter $\lambda$, which controls the balance of the fairness/adversarial loss. As shown in **Section 6.3 (Impact of the Balancing Parameter)** and **Figure 4**, an excessively large $\lambda$ causes the model to over-emphasize debiasing and lose accuracy. A moderate value (e.g., $\lambda = 5$–$7$) achieves the best trade-off, providing strong fairness (low $\Delta EO / \Delta SP$) while maintaining high predictive performance.
>
> ---
>
> **Q4:** Small issue: after Theorem 1, the explanation should not be part of the theorem.
>
> **R4:** Thank you for pointing this out. We have corrected the formatting in **Section 5.2**.

---

### Author Response · Authors · 2025-11-20
**Summary of Rebuttal Revision**

We sincerely thank all the reviewers for their efforts to review our work. In response to the valuable feedback, we have made several major updates to our manuscript, as outlined below:

1. **Enhanced Theoretical Analysis:**  We have significantly refined the theoretical section by providing a complete and rigorous proof of Theorem 1. The full derivation and logical steps are now included in the revised manuscript, with all revised content highlighted in $\color{blue}{\text{blue}}$.

2. **Additional Experiments:**  To strengthen the empirical evaluation, we conducted several new experiments and incorporated them into both the main paper and Appendix E.  These updates include:  multi-label classification results,  runtime and memory comparisons with baselines,  and an ablation study analyzing the computational cost of the AMC module.  All newly added experimental results are highlighted in $\color{red}{\text{red}}$.

3. **Improved Clarity in the Main Text:**  We have revised multiple descriptions in the main paper to improve clarity and readability, addressing the issues pointed out by the reviewers. These revised content are highlighted in $\color{purple}{\text{purple}}$. in the updated manuscript.

We hope that the revised manuscript can help address the concerns and resolve the issues raised by the reviewers.


Best,
Authors of Submission 18178

---

### Author Response · Authors · 2025-11-26
**General Response to All Reviewers**

Dear reviewers,

We sincerely appreciate your time and effort to review our paper. We are happy to see the reviewers' recognition of our paper's strengths.

You insightful suggestions are important to our paper. We have provided point-by-point responses to reviewers' comments and updated corresponding sections in our PDF. We think our responses have fully addressed your concerns — in light of this, we hope you consider **raising your score**. Please let us know in case there are any other concerns, and if so, we would be happy to respond.

Best,
Authors of Submission 18178

---

### Note · Authors · 2026-01-07

I have read and agree with the venue's withdrawal policy on behalf of myself and my co-authors.